# Holocene reconfiguration and readvance of the East Antarctic Ice Sheet

Sarah L. Greenwood [1], Lauren M. Simkins[2,3], Anna Ruth W. Halberstadt [2,4], Lindsay O. Prothro[2] & John B. Anderson[2]

How ice sheets respond to changes in their grounding line is important in understanding ice sheet vulnerability to climate and ocean changes. The interplay between regional grounding line change and potentially diverse ice flow behaviour of contributing catchments is relevant to an ice sheet's stability and resilience to change. At the last glacial maximum, marine-based ice streams in the western Ross Sea were fed by numerous catchments draining the East Antarctic Ice Sheet. Here we present geomorphological and acoustic stratigraphic evidence of ice sheet reorganisation in the South Victoria Land (SVL) sector of the western Ross Sea. The opening of a grounding line embayment unzipped ice sheet sub-sectors, enabled an ice flow direction change and triggered enhanced flow from SVL outlet glaciers. These relatively small catchments behaved independently of regional grounding line retreat, instead driving an ice sheet readvance that delivered a significant volume of ice to the ocean and was sustained for centuries.

[1] Department of Geological Sciences, Stockholm University, Stockholm 10691, Sweden. [2] Department of Earth, Environmental and Planetary Sciences, Rice University, Houston, TX 77005, USA. [3] Department of Environmental Sciences, University of Virginia, Charlottesville, VA 22904, USA. [4] Department of Geosciences, University of Massachusetts, Amherst, MA 01003, USA. Correspondence and requests for materials should be addressed to S.L.G. (email: sarah.greenwood@geo.su.se)

The East Antarctic Ice Sheet (EAIS) has undergone significant changes in its extent and behaviour throughout the Plio-Pleistocene[1–4] and there is growing recognition of its sensitivity to grounding line change during deglacial and interglacial periods[5–7] and over modern, decadal timescales[8]. Although the majority of the EAIS is grounded above sea level, approximately 19 m of sea level equivalent is stored within portions of the EAIS that rest on a bed below sea level and are susceptible to marine forcings[9]. Understanding the sensitivity and stability of EAIS catchments—for example, the magnitude of their growth and decay over long-term climate cycles, the rates of deglacial retreat and contributions to global 'meltwater pulses', and the inward-propagating influence of grounding line dynamics on fast flowing ice streams—is important in assessing the vulnerability of the EAIS to future climate and ocean changes.

The ice sheet sector of greatest change during the last glacial-interglacial cycle is arguably the Ross Sea sector, where the grounding line advanced by up to 1000 km and the catchment area expanded by ~30% during the Last Glacial Maximum (LGM) relative to the present-day extent. At the LGM, grounded ice streams filled the deep troughs that bisect the Ross Sea continental shelf[10–12]. In the western Ross Sea, ice flow was sourced from outlets of the EAIS that breached the Transantarctic Mountains[13–15], terminating at a grounding line 50–150 km inward of the continental shelf-break[11,12,16] (Fig. 1). The pronounced relief of the western Ross Sea is argued to have been a strong control on the pattern of ice flow and retreat[12], while the exact sources of ice and the configuration of flow paths in this sector have proved controversial[11,17], due to conflicting indications of ice flow trajectories around the McMurdo area based on marine till provenance analyses[14,15] and terrestrial till distribution[17,18]. Attempts to reconcile opposing flow directions in the McMurdo area call upon a reorganisation of flow in the south-western Ross Sea during deglaciation[19–21]. Recent interpretations of seafloor glacial geomorphology[20,22], terrestrial outlet glacier ice surface histories[23] and Antarctic-wide numerical ice sheet modelling[5,24] suggest a significant role of the Southern Victoria Land (SVL) sector of the EAIS in governing, and responding to, ice sheet and ice shelf dynamics in the western Ross Sea.

Fringed by the Transantarctic Mountains from the LGM grounding position to the present-day calving front, the western Ross Sea is an excellent location for examining the interplay between source outlets and regional grounding line change. How did upstream EAIS catchments respond to and interact with marine grounded ice flow, grounding line forcings, and the establishment and loss of the floating Ross Ice Shelf? Here we integrate newly acquired geophysical seafloor data with >20 years' legacy multibeam data from the western Ross Sea ('Methods'; Supplementary Fig. 1), and analyse the geomorphological footprint of EAIS deglaciation. We find geomorphic and stratigraphic evidence for complex palaeo-ice flow behaviour in the western Ross Sea, closely tied to EAIS catchments. A non-uniform pattern of deglaciation comprises, in particular: reversal of ice flow direction in southern Drygalski Trough; a retreat and readvance event in southern JOIDES Trough linked directly to Southern Victoria Land catchments, with a spatial scale >50 km and sustained discharge to the ocean over decades to centuries; and grounding line oscillations or reconfiguration of a remnant Crary Bank ice rise. Compiled marine and terrestrial chronological data place these events in the early-mid Holocene. We interpret a reconfiguration of the SVL sector of the EAIS in response to the removal of the regionally grounded ice sheet from the Ross Sea, permitting the unzipping of ice sheet sub-sectors that yield independent dynamic responses. Our findings from the western Ross Sea are among a spate of recent studies that find

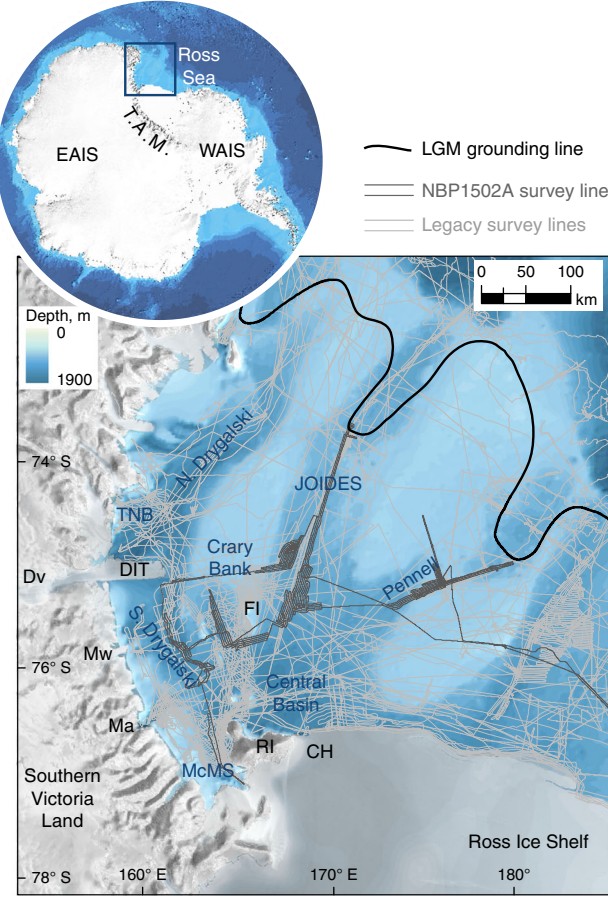

**Fig. 1** The western Ross Sea. Multibeam data were compiled from NBP cruises 1994–2015, Oden cruises 2007–2010 and Araon cruises 2013–2015. LGM grounding line from ref. [12]. Bathymetry and inset panel image from IBCSO[76]. TNB = Terra Nova Bay, Dv = David Glacier, DIT = Drygalski Ice Tongue, Mw = Mawson Glacier, Ma = Mackay Glacier, McMS = McMurdo Sound, RI = Ross Island, FI = Franklin Island, CH = Coulman High, T.A.M. = Transantarctic Mountains, EAIS = East Antarctic Ice Sheet, WAIS = West Antarctic Ice Sheet

reorganisations in different sectors of the Antarctic ice sheet following retreat from the LGM[23,25–28]. These collectively raise questions of the vulnerability, stability or resilience of different ice sheet sectors to grounding line change, and demand analysis of why different sectors may behave in different ways including, in cases such as we present here, advance for a period of centuries in the face of climate warming and sea level rise.

## Results

**Glacial landforms.** A well-developed suite of glacial landforms is observed on the seafloor throughout the western Ross Sea (Figs. 2–5), within which distinct assemblages are identified that relate to ice flow paths and margin positions during the last deglacial period. Approximately 5965 glacial lineations are mapped, ranging from small fluting across the crests of grounding line landforms (c. 200 m length; e.g., Fig. 2c) to highly elongate lineations in northern Drygalski Trough (up to c. 13,250 m length; e.g., Fig. 2b) that record focussed ice stream flow. A major lineation assemblage sweeps apparently continuously around southern Crary Bank (Fig. 2a), flowing southward out of southern Drygalski Trough and ultimately NE into southern JOIDES; in JOIDES Trough itself lineations are sparse. A distinct group of lineations are oriented NNW off the eastern coast of Ross Island.

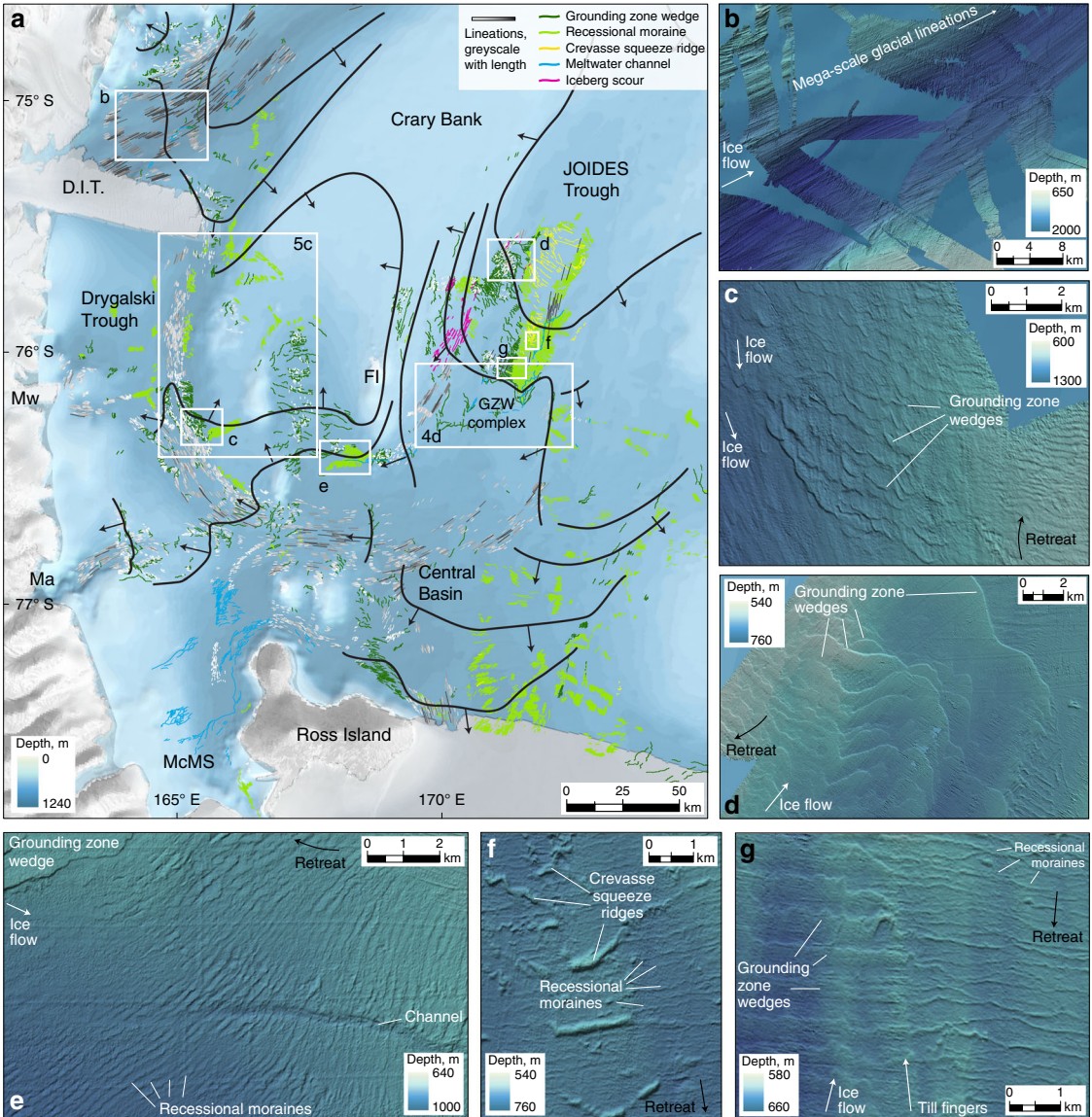

**Fig. 2** Glacial landform assemblages in the western Ross Sea. **a** Generalised retreat pattern (heavy black, retreat in direction of black arrows), based on mapped glacial landforms: lineations (greyscale: white to black with increasing length), grounding zone wedges (dark green), moraines (light green), crevasse squeeze ridges (yellow), meltwater channels (blue), iceberg scours (pink). Background bathymetry from IBCSO[76]. **b**–**f**. Landform examples, with locations marked on **a**, and multibeam data compiled and visualised as described in the 'Methods'. **b** Mega-scale glacial lineations emanate from David Glacier and flow NE through northern Drygalski Trough. **c** Small-scale grounding zone wedges with fluted crests overprint southward-directed lineations in southern Drygalski Trough, and mark retreat from the trough onto Crary Bank to the NE. **d** Stacked grounding zone wedges on the western flank of JOIDES Trough, pinned on Crary Bank. **e** Recessional moraines south of Franklin Island mark grounding line retreat to the west. **f** Irregular ridges within a recessional moraine field are interpreted as basal crevasse fill. **g** Lateral transition from small-scale grounding zone wedges to recessional moraines across a single, retreating grounding line. These overlie broad, subtle, N-S oriented till ridges or 'fingers'

A small group of S-N oriented drumlins are seen in the deepest part of McMurdo Sound while a WNW-ESE field rounds the tip of Cape Bird; the direction of ice flow recorded by either of these sets is difficult to conclusively determine[20].

Grounding line landforms dominate the seafloor of the western Ross Sea, recording ice margin retreat. Recessional moraines and grounding zone wedges (GZWs) are arranged in a dense field in southern JOIDES Trough, in the forefield of an intermediate-scale[12] composite GZW complex (Fig. 2a, d, f, g). Small-scale GZWs < 10 m in amplitude are stacked upon each other on the flanks of Crary Bank (Fig. 2d), while lateral transitions from asymmetric wedge to symmetric moraine morphologies are also observed (Fig. 2g). Our newly acquired, high resolution bathymetric data (cruise NBP1502A) permit us to interpret far more

extensive fields of recessional moraines (e.g., Fig. 2e) than had previously been observed in older, poorer quality data, in particular over Crary Bank (Fig. 5) and throughout Central Basin. A group of irregular, criss-crossing ridges with highly variable crest relief (2.5–25 m) are observed amid a field of recessional moraines (typically <5 m) in southern JOIDES Trough (Fig. 2f). These ridges are aligned generally oblique to the grounding line landforms, though they show occasional spatial transition to or connection with the moraines. While their arrangement does not form the regular, rhombohedral pattern considered typical of basal crevasse squeeze ridges[29,30], we nonetheless interpret these as such, formed directly behind the grounding line and indicating the possibility for full-thickness calving.

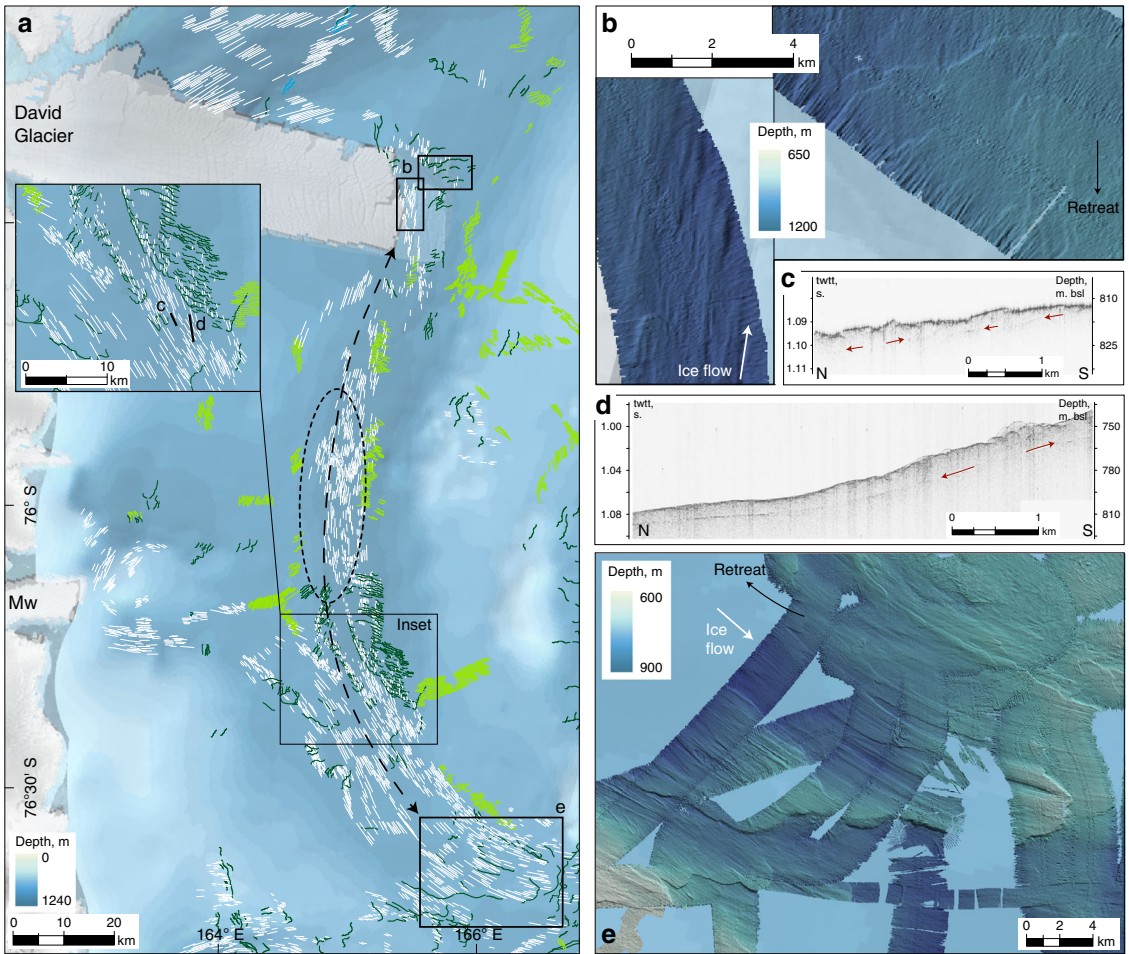

**Fig. 3** Flow reversal in southern Drygalski Trough. Background bathymetry from IBCSO[76]; acoustic data compiled and visualised as described in the 'Methods'. **a** Mapped lineations (white) form an apparently continuous flowset south of the Drygalski Ice Tongue. Relationship with grounding line landforms (green) suggests, however, that lineations flow northward at the northern end of the group (panel **b**, white arrow in direction of flow), and southward at the southern end (panel **e**, white arrow in direction of flow). **c**, **d** Subbottom acoustic profiles from NBP1502 show fragments of a buried till horizon, only observed in the southern part of the trough. Increased data coverage from coastal waters (e.g., around 164° E) would greatly improve understanding of the role of SVL outlets in western Ross Sea palaeo-flow structure and deglacial grounding line dynamics

**General retreat pattern**. Figure 2a presents the general pattern of ice retreat in the western Ross Sea, which is broadly consistent with recent interpretations[12,22]. Retreat from northern Drygalski Trough proceeded southward and westward towards David Glacier and the Terra Nova Bay outlets, while small recessional moraines on the NW flank of Crary Bank mark grounding line retreat onto the bank and the partial separation of ice flowing out of the Transantarctic Mountains and off the bank (Fig. 2a). The pattern of retreat in JOIDES Trough partially mirrors this behaviour. The grounding line in the central part of the trough retreated steadily southward along the trough axis, depositing regularly spaced small-scale GZWs and moraines, while simultaneously pinned on the eastern flank of Crary Bank where GZWs are stacked upon each other (Fig. 2d). Basal crevasse squeeze ridges extend obliquely across the moraines from the suture with stacked GZWs (Fig. 2d, f), suggesting these crevasses are linked to the unzipping of ice draining through the trough from ice flowing off the bank, and the consequent opening of a deep grounding line embayment within JOIDES Trough. Retreat from the intermediate-scale GZW complex in southern JOIDES Trough proceeded south and westward, as southward retreating ice in the Central Basin separated from a major ice lobe curving around Crary Bank. The westward retreating flow separated into

component catchments: back-stepping GZWs, moraines and glacial lineations mark retreat into Mackay Glacier, southern Drygalski Trough and Mawson Glacier, while moraines step back onto Crary Bank from all directions (e.g., Figs. 2c, d and 5). This indicates that a semi-independent ice rise established on the bank top in the late stages of deglaciation.

Large-scale westward retreat of grounded ice within the western Ross Sea indicates that SVL outlets of the EAIS were an integral component of the deglacial flow dynamics[22]. Within the general retreat pattern presented in Fig. 2a, we find several instances of flow reconfiguration linked to the behaviour of SVL outlets: flow reversal in southern Drygalski Trough; a grounding line readvance event in southern JOIDES Trough; and reconfiguration of the remnant Crary Bank ice rise.

**Flow reversal in southern Drygalski Trough**. Glacial lineations aligned with the trough axis are found throughout southern Drygalski Trough (Figs. 2a and 3). In southernmost Drygalski Trough, lineations with northwest-southeast orientations lie on top of GZWs that record southward ice flow and northward grounding line retreat (Fig. 3e). Adjacent to Drygalski Ice Tongue, the same landform relationship indicates northward ice flow and southward retreat (Fig. 3b), while north of the ice tongue

northward drainage of David Glacier dominates the landform record (Fig. 2b). In the central trough axis, in front of Mawson Glacier, lineations oriented north–south but of indeterminable direction (Fig. 3a, circled) link these two sets in an apparently continuous lineation assemblage. However, this is a false impression. If this central group of lineations are directed northward, their source must lie at least as far south as Mackay Glacier; if the central group are directed southward, then their source must lie at least as far north as David Glacier. The trough-aligned orientation and position of these lineations in the central part of southern Drygalski Trough precludes a scenario of simultaneous northward and southward flow diverging from SVL outlet sources. The opposing flow directions at either end of the Drygalski assemblage must therefore represent two opposing, non-contemporaneous flow events, and imply an intervening shift (reversal) in flow direction of SVL outlet glaciers.

The dominant retreat assemblage in southern Drygalski Trough is clearly associated with southward flow (northward retreat; Fig. 3a, e). This assemblage backsteps into the troughs of Transantarctic Mountain outlets such as Mackay and Mawson Glaciers, as well as unzipping through the deepest parts of southern Drygalski Trough with a component retreating onto Crary Bank. Furthermore, in the southernmost part of Drygalski Trough our acoustic stratigraphy reveals fragments of an horizon buried 5–7 ms twtt (~4.5 m) below an acoustically transparent unit interpreted as the uppermost till (Fig. 3c, d); this buried horizon shallows or is absent further north towards the Drygalski Ice Tongue. We suggest that the southward flow (northward retreat) is the final flow configuration, likely responsible for depositing a thick till unit in southernmost Drygalski Trough during retreat. This was preceded by a phase of northward flow, potentially associated with a till unit that is buried in the south but forms the surface unit further north. During deglaciation, therefore, the direction of flow through southern Drygalski Trough switched, accompanied by enhanced contributions of the SVL outlet glaciers.

**Readvance into southern JOIDES Trough**. Grounding line landform assemblages dominate southern JOIDES Trough (Figs. 2a and 4). GZWs and moraines, oriented perpendicular to the trough axis and spaced regularly at 200–600 m (e.g., Fig. 2g), backstep over a distance of ~80 km to where an intermediate-scale GZW[12] buries a southward continuation of the moraine field (Fig. 4b, c). The buried moraines require that the grounding line had stepped southward of the GZW position, prior to readvancing across the moraine field.

The readvance GZW is supplied by ice flow corresponding to the main set of glacial lineations that sweeps around southern Crary Bank (Fig. 4a, c—flowset *fs1*). In the western part of the readvance grounding line, close to the flank of Crary Bank, glacial lineations transition across the grounding zone into subparallel and curvilinear iceberg furrows (Fig. 4a). Towards the east, the lineations dissipate beyond the GZW. This lineation set contrasts with an older group in JOIDES Trough, which underlies the mid-JOIDES recessional moraines (e.g., till 'fingers' in Fig. 2g) and therefore represents flow pre-dating the initial retreat. This early flow direction is ~10° east of N and suggests a component of flow from Central Basin and the Ross Sea interior, in contrast to the E-to-NE readvance assemblage that points to SVL outlets as the main source of ice responsible for this later event. The retreat-readvance sequence therefore accompanied a reconfiguration of the ice flow structure with dominant flow from the SVL sector during readvance.

The GZW and readvance lineations clearly superimpose W-E oriented recessional moraines (Fig. 4b) up to 50 km south of the GZW front, constraining the minimum distance of readvance to ~50 km (Fig. 4c). The recessional moraine field continues in consistent clusters even further south, recording unzipping of ice through Central Basin as the grounding line backstepped into increasingly deep waters (Fig. 4c). It is possible, therefore, that the entire moraine field, up to the limits of available data just ~10 km from the present ice shelf calving front, records an early phase of grounding line retreat that was followed by significant readvance of ice draining from SVL. This would extend the magnitude of the readvance event to 100–150 km, and it would imply that the earlier retreat proceeded from the LGM position on the mid-outer continental shelf[12] over ~225 km to at least the present calving line. Patchy, north/north-westward lineations east of Ross Island (Figs. 2a and 4c—flowset *fs3*) would also have been truncated by the SVL readvance set, under this interpretation. However, while the moraines appear to be a continuation of the same field as those that are overprinted, no readvance landforms directly superimpose the southernmost landform assemblages, and therefore evidence for a much greater magnitude readvance remains equivocal. The seafloor deepens here and it is possible that a readvance lobe that grounded in southern JOIDES Trough remained afloat in this deeper zone. On the other hand, continuity of meltwater channel systems in front of and behind the GZW complex[31] (Figs. 2a and 4b) could argue against such a large-scale reconfiguration. For the purposes of quantifying the event, we conservatively limit our estimate of the readvance distance to a minimum of 50 km.

Unfortunately, the western Ross Sea currently lacks a sufficiently robust chronology[11] to constrain the timing of the readvance and, importantly, the rates of grounding line migration that are implicated. Instead, we use our geophysical data and reconstruction of the event's spatial magnitude to place upper and lower bounds on the event's duration. The readvance event terminated with a sufficiently long stillstand to accumulate a 20 m high, ~3 km$^3$ sediment wedge (GZW1), the most distal wedge within a large grounding zone complex. Our acoustic stratigraphy reveals a more-or-less continuous buried surface beneath the upper, acoustically transparent unit that comprises the wedge complex (Fig. 4e, f). This surface extends underneath the main GZW1 as well as two further GZWs to its south (1b & 2), and allows us to explore the sediment distribution within the whole assemblage ('Methods'; Fig. 4d). With a mean thickness of $3.16 \pm 2.5$ m (1 $\sigma$), the unit thickness peaks at 19.5 m and has a typical thickness at the GZW1 topset break of 10–14 m. The unit thins to the south although, behind the southernmost GZW2, it maintains a thickness of ~1.4 m. Given the total volume of sediment accumulated in GZW1 and using a range of reported subglacial sediment fluxes that encompass outstanding uncertainties in transport rate and mechanisms (order $10^2$–$10^3$ m$^3$ a$^{-1}$ m$^{-1}$; 'Methods'), we calculate an upper bound of c. 1000 years on the duration of the stillstand at GZW1, and a lower bound as short as c. 80 years; the duration of the whole grounding zone complex is c. 100–1600 years (Supplementary Table 1, Supplementary Fig. 2). We find, therefore, that this readvance was a sustained event.

A readvance of ~50 km would propel ~3000–4000 km$^3$ of ice to the advanced grounding line while, with bounds of c. 100–1600 years and a typical ice flow velocity of 200–800 m a$^{-1}$ (refs. [32,33]), we bracket a likely volume of ice discharged through the event's maximum position (Fig. 4g) to the order of ~3500 to >60,000 km$^3$, or ~15–100 Gt per year ('Methods'; Supplementary Table 2). This grounding line discharge exceeds that of the modern David Glacier (~8 Gt a$^{-1}$; ref. [34]); the upper bound is comparable to discharge from today's major outlets such as Totten or Pine Island Glaciers (~80–110 Gt a$^{-1}$)[34]. This readvance event not only accompanied a reconfiguration of ice flow sourced from the

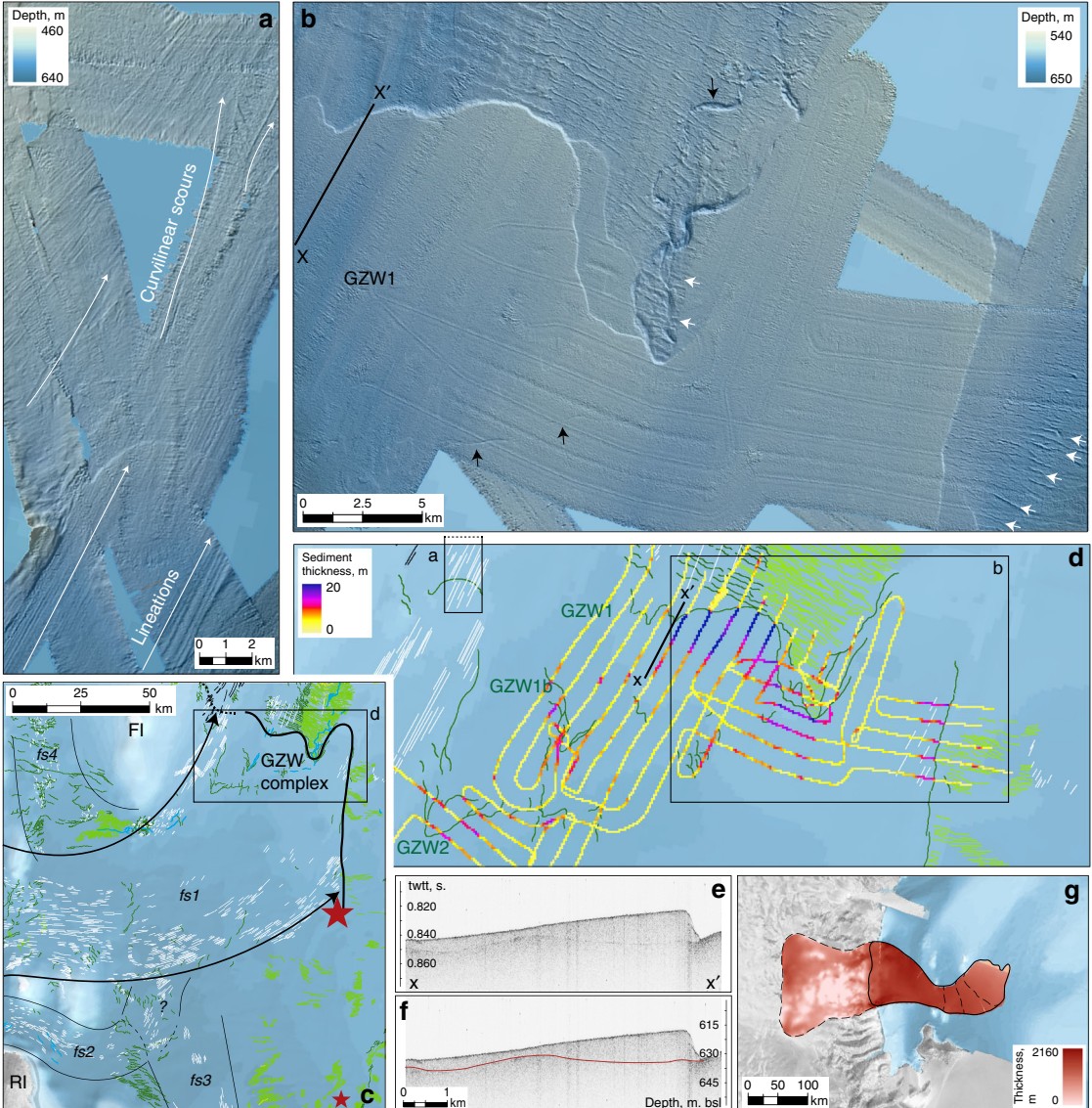

**Fig. 4** Readvance assemblage in southern JOIDES Trough. Background bathymetry from IBCSO[76]; acoustic data compiled and visualised as described in the 'Methods'. **a** Lineations crossing a grounding zone wedge front transition to curvilinear iceberg/keel scours distally. **b** Large, composite GZW (1) overprints an earlier retreat sequence comprising W-E recessional moraines (arrowed in white). A meltwater channel system both under- and overlies GZW1 (arrowed in black). **c** Lineation assemblage (in white) indicates flow from Southern Victoria Land to the JOIDES GZW complex (flowset *fs1*), possibly joined by *fs2* and apparently truncating *fs3*, indicating unzipping of flow in Central Basin prior and/or subsequent to readvance. Large red star marks the southern limit of where landforms associated with the readvance superimpose earlier retreat moraines, i.e. the minimum distance of initial retreat, prior to readvance; smaller star marks southern continuation of the moraine field. FI = Franklin Island, RI = Ross Island. **d** Sediment thickness within the GZW complex (GZWs 1, 1b & 2), based on a buried horizon (**e**, **f**) mapped from the NBP1502A subbottom acoustic survey. **g** Potential catchment of the readvance event, assuming a range of durations and flow velocities (see 'Methods')

SVL sector of the EAIS, but contributed a potentially significant volume of ice to the ocean.

**Reconfigurations of Crary Bank Ice Rise**. Following the southern JOIDES readvance event, lineations and grounding line landform assemblages clearly document retreat to Mackay and Mawson Glaciers on the SVL coast, and onto Crary Bank. Grounding line landforms retreat up onto Crary Bank from all sides (Figs. 2 and 5a, c–e). In southern Drygalski Trough, opposing sets of small-scale grounding zone wedges record the separation of ice retreating westward towards Mawson Glacier, and north-eastward onto Crary Bank (Fig. 5c, d). A bank top ice

rise therefore developed, at least semi-independent of the main EAIS.

Recessional moraines atop and encircling Crary Bank cannot, however, be linked together to map out a concentric pattern of retreat (Fig. 5c). Rather, small-scale lineations and grounding line landforms on the western part of the bank, recording east-to-west margin retreat (Fig. 5b, c—light blue set), conflict with sets of grounding line landforms that record trough-to-bank retreat from both the west and the south (Fig. 5a, c–e—green set). Furthermore, moraines that flank Crary Bank along much of southern Drygalski Trough (Fig. 5c—red set) are difficult to reconcile with either of these other groups. In the absence of superimposed landforms, and while multibeam and acoustic stratigraphic data coverage remains fragmentary in this area, we

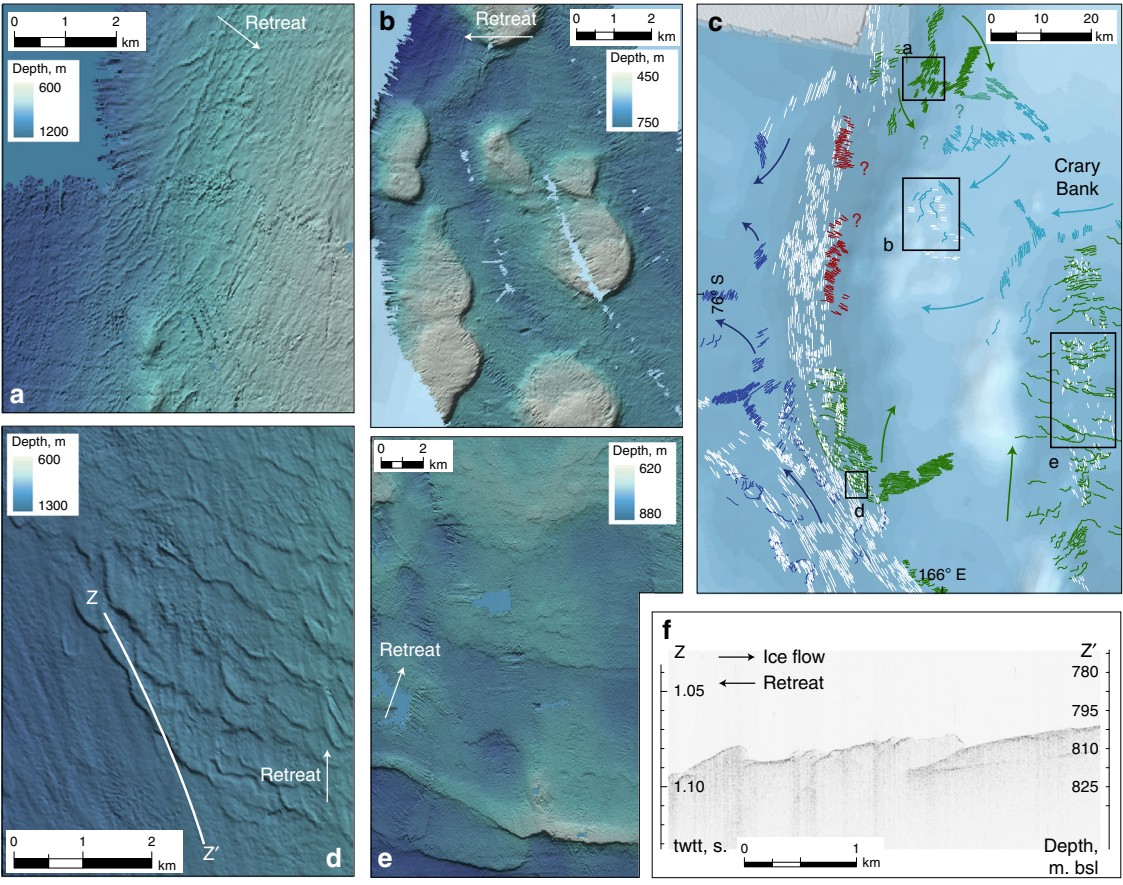

**Fig. 5** Retreat landforms associated with a semi-independent Crary Bank ice rise. Background bathymetry from IBCSO[76]; acoustic data compiled and visualised as described in the 'Methods'. Recessional moraines (**a**) and GZWs (**d, e**, Fig. 2d) record trough-to-bank retreat from the NW, SW, S, E. **b** Wedges pinned on seamounts record bank-to-trough retreat. **c** Three separate clusters of retreat landforms (blue, red, green; retreat directions for each set arrowed) cannot be reconciled with a single ice rise configuration. **f** Subbottom acoustic profile shows underlying till unit cannibalised by margin oscillation associated with small-scale grounding zone wedges in **d**

cannot yet resolve the pattern or relative timing of final grounding line retreat on Crary Bank. Multiple conflicting sets of retreat landforms, however, must indicate reconfiguration(s) of the remnant ice rise grounding line.

Glacial lineations and landforms representing east-to-west grounding line movement graze the western flank of Crary Bank (Fig. 5b, c—light blue set) yet are absent from deeper waters in Drygalski Trough, appearing again as the topography shallows westwards towards the SVL coast (Fig. 5c—dark blue set). This distribution suggests an episode in which SVL outlets fed bank-top ice while remaining afloat across the deepest part of the trough (i.e., a re-grounded ice shelf). It follows that this configuration may post-date ungrounding of the retreating ice in Drygalski Trough, and that the Crary Bank margin shifted in geometry in conjunction with the dual trends of retreat towards source and thinning towards flotation over deep waters.

**Chronology**. We identify several instances of ice flow and grounding line reconfiguration during the deglaciation of the SW Ross Sea. Relative to one another, we infer that the southern Drygalski flow switch is dynamically linked to the JOIDES grounding line readvance, which is clearly supplied by the younger, south and eastward flow from Mackay and Mawson outlets; reconfiguration of Crary Bank ice rise would have occurred during (a) final phase(s) with a restricted grounding line extent. The absolute timing of these events is, unfortunately, difficult to constrain, despite much previous work into western

Ross Sea deglaciation sequences[11] and renewed attempts to target specific grounding line features for dating. Carbonate preservation is extremely poor here, and radiocarbon dating using the acid insoluble organic fraction of bulk sediments is widely considered to be unreliable in dominantly terrigenous glaciomarine sediments, poorly constraining the timing of grounding line retreat[11,35,36]: published open marine onset dates in our area of interest range from ~4.2 to 12.6 $^{14}$C ka, a window that does little to constrain the grounding line dynamics we report. A carbonate age from Coulman High, off the east of Ross Island, gives a minimum of 8.6 cal ka for grounding line retreat in the Central Basin[36]. Our evidence for (at least partial) deglaciation of Central Basin prior to SVL readvance raises the possibility that this age may pre-date the reconfiguration. Indeed, the Coulman High core lithostratigraphy comprises a short period of open water conditions, from which the carbonate sample was recovered, prior to a regrowth of ice shelf cover[36]. The carbonate age therefore indicates either that the whole retreat-readvance-final retreat sequence took place prior to 8.6 cal ka, and the Coulman High lithostratigraphy represents a later calving line oscillation that our grounded ice reconstruction does not capture; or that the first retreat of both the grounding and calving line progressed at least as far as Coulman High, and that the grounded ice readvance we reconstruct around southern Crary Bank was accompanied by ice shelf re-growth over the deep waters, shortly after 8.6 cal ka.

Terrestrial ages from the SVL coast constrain the final deglaciation of the western Ross Sea, and are therefore minimum limiting ages on all of the events we report here. Raised shoreline

ages north of Drygalski Ice Tongue suggest this coast was at least seasonally free of ice (i.e., the calving line had retreated) by 8 cal ka[37,38]. To the south, the grounding line retreated into Mackay Glacier trough at 6.8–6.0 $^{10}$Be ka[23] and seasonally open marine conditions began ~7.5–6 cal ka[16,19,39,40]. Within the available window of time, we tentatively suggest that the readvance of SVL ice into southern JOIDES Trough occurred sometime around 8.5 cal ka; the advanced grounding line was sustained for a few centuries; and the ensuing retreat of both the grounding and calving lines was accomplished within ~1500–2000 years. This time window is consistent with sediment flux-based estimates of GZW formation time, but a refined chronology is nonetheless imperative to better constrain the scale and rate of ice margin change and ice margin stability, and to refine understanding of sediment fluxes and GZW formation processes. The Crary Bank Ice Rise was the final remnant of grounded ice in the western Ross Sea.

## Discussion

The deglaciation of the Ross Sea has proven controversial[11,17], in terms of its timing (pre-global LGM or post-13 ka[41–43]), style ('swinging gate' or 'saloon door'[44,45]) and the controls governing ice retreat. Halberstadt et al.[12] propose a pattern of deglaciation characterised by complex bank-trough relationships and embayment-driven 'unzipping' of different sub-sectors of the formerly grounded ice sheet in the Ross Sea Embayment, in a heterogenic fashion. We find the pattern of retreat shown by glacial landforms in the western Ross Sea entirely consistent with this style, and in keeping with growing recognition that this sector experienced shifts in its ice flow configuration[19–21], that the SVL outlet glaciers are sensitive to offshore dynamics[23], that coastal and deeper marine deglaciation may be decoupled[36] and that this sector's deglaciation is decoupled from general southward grounding line retreat into the southern Ross Sea embayment[12,22,32]. Encouragingly, recent numerical ice sheet models[5,7] echo this behaviour, demonstrating unzipping of grounding line embayments, ephemeral flow direction shifts (notably of David Glacier), retreat-readvance events and the development of residual bank-top ice caps or ice rises.

We have found traces of ice sheet reorganisation during deglaciation of the western Ross Sea, associated with at least one readvance that delivered a significant and sustained volume of ice to the ocean. We link this readvance to relatively enhanced ice flux from the outlet glaciers of SVL and the local reversal of flow direction witnessed in southern Drygalski Trough. These observations couple ice flow behaviour with dynamics of grounding line change, and it is therefore important to consider what mechanisms triggered grounding line retreat and readvance, and drove SVL outlets to dominate flow geometry late in the deglaciation of the western Ross Sea.

Grounding zone self-stabilisation due to sediment progradation may present a conservative explanation of our observations of moraine burial: that a stillstand in the initial retreat enabled a period of GZW growth, such that the grounding line became less sensitive to buoyant forces and stabilised[46] with ensuing progradation of a large wedge. However, this would require progradation of a wedge at least 50 km downstream of its seeding location, a distance towards the upper limit of previously reported GZWs[47]. This scenario would also need to explain the initial stillstand that would have allowed a GZW to seed. This stillstand position would be located where the reverse bedslope had in fact steepened after the initial retreat, and where JOIDES Trough widens into the Central Basin (Fig. 6); neither setting is conducive to grounding line stabilisation[48]. Finally, while a stillstand and readvance could be encouraged by either a change in ocean

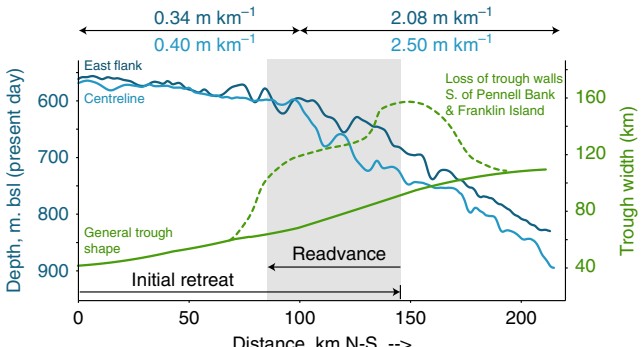

**Fig. 6** JOIDES Trough geometry through which grounding line advance occurs. Initial retreat stabilisation and grounding line advance through GZW progradation appears an unlikely explanation, since the retreated stable position would occur where bed slope (in blue) increases and where trough width (in green) widens. An additional dynamic component is therefore required to drive a readvance

forcing at the grounding line (reduced submarine melting[32,49–51]), or a buoyancy threshold feedback with isostatic rebound[25,52,53], none of these mechanisms logically explains why a grounding line advance was accompanied by a flow reconfiguration, rather than simply regrowth that maintained the initial flow geometry.

The retreating grounding line in JOIDES Trough encountered a steadily deeper and wider physiographic setting, and therefore we may expect unstable grounding line behaviour[48,54]. Basal crevasse squeeze ridges within the recessional moraine field of JOIDES Trough (Fig. 2f) raise the possibility of deep draught crevassing and calving close to the grounding line during an initial phase of retreat, while the shape and distribution of recessional moraines and GZWs reveal the development of a grounding line embayment. The opening of an embayment divides a catchment into its component sub-sectors that can thereby develop independent dynamics. We suggest that an embayment in Central Basin, pinned on Crary Bank to the west (Fig. 2d) and ultimately also Ross Island and Ross Bank, would allow for the unzipping of different ice sheet sub-sectors that had fed flow to the western Ross Sea.

Two complementary processes likely enhance flow through SVL and trigger flow reorganisation. The loss of grounded ice buttressing would stimulate flow acceleration, while the possibility that initial grounding line retreat was also accompanied by loss of an ice shelf would further enhance this effect[55,56]. Additionally, the increase in moisture availability from an open sea may have further nourished nearby outlets. The Coulman High core stratigraphy[36] points to a period of open marine conditions in the Central Basin followed by renewed ice shelf cover, consistent with suggestions of two-stage ice shelf retreat[57]. Vacation of grounded and, potentially, shelf ice from Central Basin enabled SVL outlet glaciers to freely expand and accelerate, stimulating S and E-ward flow from Drygalski Trough (Fig. 7a) in a response sustained for a few centuries.

Embayment and readvance behaviour is consistent in style, if not in timing, with modelled Ross Sea deglaciation[5,7]. Model outputs show that a broad drainage basin with low ice surface profile develops in Central Basin once the grounding line retreats as far as southern JOIDES Trough. Limited lateral drag through a broad basin would encourage enhanced and preferential discharge of SVL outlets. We note that only minimal upstream thinning of one of the SVL outlets, Mackay Glacier, is documented to have occurred at this time[23], and suggest that enhanced accumulation may have offset surface lowering in the terrestrial portion.

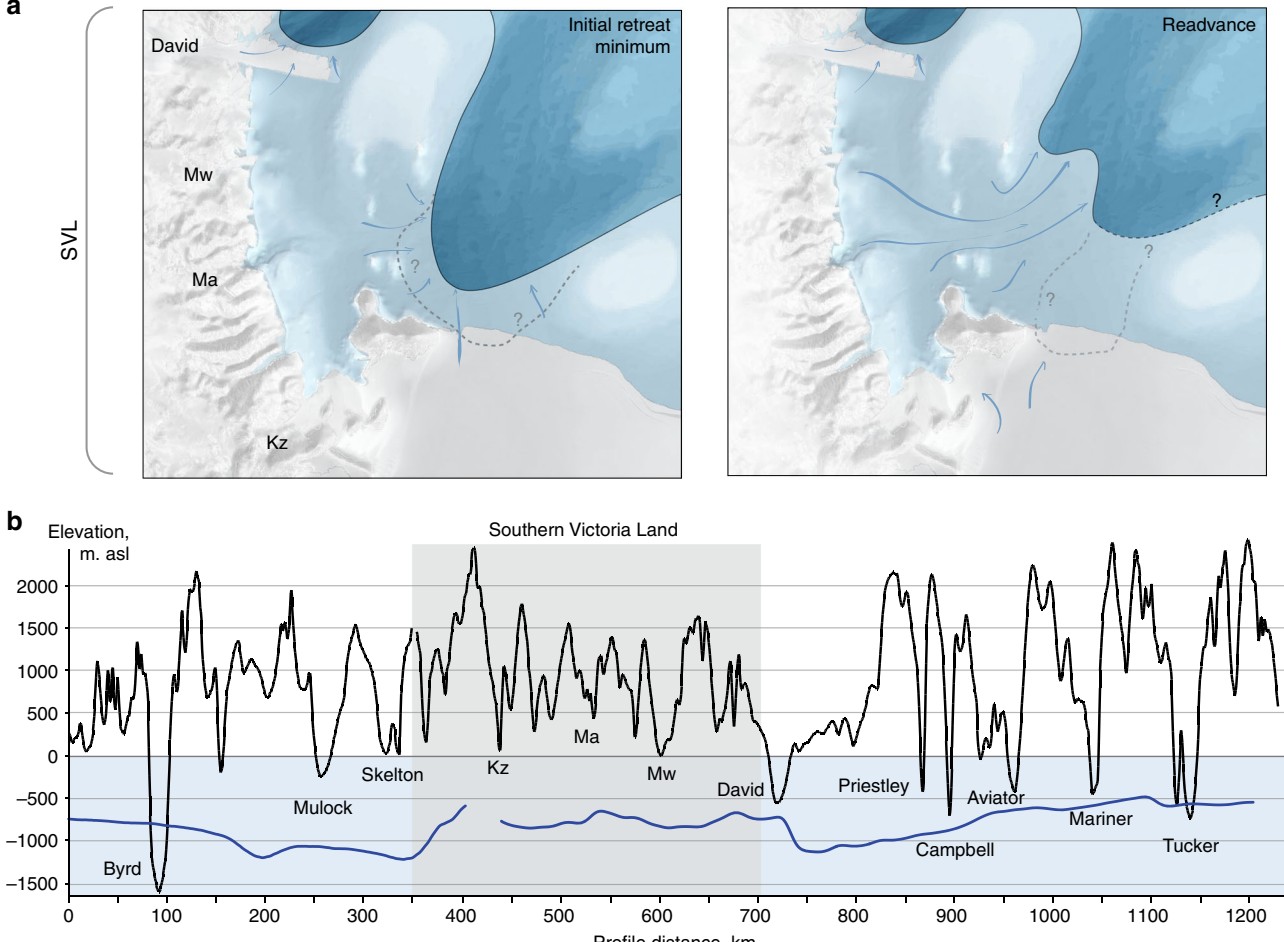

**Fig. 7** Readvance configuration and relationship to the relief of contributing glacier outlets. **a** Initial retreat of ice through JOIDES Trough creates embayment in Central Basin, allowing SVL and southern TAM subsectors to unzip and behave independently. SVL outlets drive a readvance into southern JOIDES Trough. The magnitude of retreat prior to readvance is uncertain, as is the grounding line / ice shelf extent over Central Basin during readvance (dashed lines). Background bathymetry from IBCSO[76]. **b** Topographic and bathymetric profiles of the TAM/Ross Sea coast (from IBCSO[76]); TAM profile (black) follows coast approximately 50% to watershed; Ross Sea profile (blue) follows coastal trough axis. Outlet glaciers of SVL are considerably less incised than either the northern or southern TAM and therefore ice entering the Ross Sea coastal trough from SVL is thinner. This sector will therefore be more vulnerable to drawdown, acceleration (readvance) and grounding line instability (subsequent retreat), until the grounding line retreats to and stabilises at the coastal zone. Kz = Koettlitz Glacier, Ma = Mackay Glacier, Mw = Mawson Glacier

The geometry of the contributing valleys through the Transantarctic Mountains is also likely relevant here. Since SVL troughs are not deeply excavated (Fig. 7b), ice draining through these outlets is thin relative to that draining through the overdeepened Transantarctic valleys both to the north and further to the south. Where these shallow outlets discharge into the Ross Sea and encounter a deep coastal trough, the thinner ice and lower mass flux will make the downstream grounding line more vulnerable to flotation, compared to where thick ice from deeply cut outlet valleys feeds the grounding line. We suggest that as the regional grounding line approached these thinner EAIS outlets in SVL, this ice thickness vulnerability manifested as a time-limited dynamic response. A combination of embayment-driven isolation of ice sheet sub-sectors, and a source catchment delivering thin ice into the southwestern Ross Sea, primed this sector for a low profile readvance prior to eventual grounding line retreat towards the coastal zone.

Ice sheet reconfiguration in the SVL sector of the EAIS during the last deglaciation is one of a suite of Holocene age reorganisations recently identified around the Antarctic ice sheet[23,25–28]. These collectively highlight contrasts in the vulnerability or resilience of different ice sheet sectors and raise questions of why different sectors behave differently. While the timing, topographic, glaciological and oceanographic contexts vary between these cases, dynamic changes in ice flow linked to grounding line behaviour are common to all. We suggest that as the grounding line encounters the more complex topography commonly found approaching the continental interior, changes in grounding line shape and differences in source catchment geometry trigger responses in ice flow configuration and behaviour. Here we show that, consistent with the style of deglaciation across the wider Ross Sea, development of an embayment in the grounding line allowed contributing sub-sectors of the ice sheet to unzip and jostle. Drawdown and enhanced flux through the SVL outlets drove local flow reorganisation and an ice margin readvance that was sustained for a few centuries and delivered a significant volume of ice to the ocean. The flow behaviour of sub-catchments may therefore be independent of, and in this case opposite to, regional grounding line change. We additionally suggest a feedback with long-term patterns of outlet glacier erosion whereby shallowly excavated valleys that debouche into a deep marine setting are more vulnerable to transient drawdown and dynamic

distal grounding line behaviour than their deeply excavated counterparts.

## Methods

**Acoustic data collection and processing.** New multibeam echo-sounding data were collected on cruise NBP1502A, aboard the RVIB *Nathaniel B. Palmer*, with a Kongsberg EM122 operating in dual swath mode, at 12 kHz frequency with a 1° × 1° beam width. The swath angular coverage was set to 62° × 62° with an overlap between survey lines typically ~30–60%. Sound velocity control was achieved with regular Expendable Bathy Thermograph casts. Data were processed in Caris v8, ping-edited on board and gridded using the CUBE algorithm to 20 × 20 m in the UTM zone 58-south projection.

The RVIB *Nathaniel B. Palmer* has collected multibeam data from the western Ross Sea since 1994. These legacy data (Supplementary Table 3) were retrieved from the Marine Geoscience Data System archive at www.marine-geo.org, re-processed and gridded together. Kongsberg EM120 data from 2003–2013 were imported into Caris, maintaining any original ping-editing in the archived files and performing further cleaning of the raw data. Single grids of these combined datasets were built with a horizontal resolution of 30–35 m, depending on data quality and density. Data collected prior to 2003 (SeaBeam system) were gridded without further processing at 30–40 m. These data were complemented with transit survey lines from three RVIB *Oden* cruises, which collected data with a Kongsberg EM120 (OSO0708) and EM122 (OSO0910, OSO1011; http://oden.geo.su.se), and EM122 survey data from RV Araon (2013 and 2015; www.marine-geo.org). All surfaces were gridded in UTM58S projection (Supplementary Fig. 1).

Multibeam surfaces were visualised using a variety of hillshades and colour ramps in Caris and in ArcGIS. Seafloor morphology was interpreted in both software environments, and mapped in ArcGIS.

Subbottom acoustic data were collected during NBP1502A with a Knudsen CHIRP 3260 system, using a frequency of 3.5 kHz and a 0.25 ms pulse width. These data were imported into IHS Kingdom software for visualisation and analysis, and seafloor depths converted from two-way travel time using a sound velocity of 1500 m s⁻¹.

**Readvance event: sediment delivery, duration, ice volume.** Magnitudes and rates of grounding line change are important in defining grounding line stability or vulnerability. Unfortunately, difficulties of radiocarbon dating in the marine sectors of the western Ross Sea[11,35,36] mean that we currently lack the means to constrain the timing of grounding line readvance in the Central Basin and southern JOIDES Trough. We therefore exploit our geophysical data to place limits on the event's duration, using estimates of sediment transport to the grounding line. Our approach, similar to that adopted in other studies[58–61], is designed to encapsulate the range of uncertainties in sediment transport mechanisms and rates, and express these as upper and lower bounds on our reconstruction of the readvance event's magnitude.

We calculate event duration (*t*, in years) as the volume of sediment deposited (*v*, in cubic metres) per metre of grounding line divided by the annual sediment volume flux (*Q*, in cubic metres per year) over the same unit width, i.e.,

$$t = \frac{v}{Q} \qquad (1)$$

A regular net of NBP1502A survey lines, totalling ~700 km, crosses the grounding zone wedge (GZW) complex in southern JOIDES Trough. Acoustic stratigraphic reflection horizons corresponding to the seafloor and to a widely identifiable subbottom reflector defining the base of the GZWs were mapped in IHS Kingdom. The two-way travel time (TWTT) difference between the two horizons defines the thickness of the GZW and reveals its spatial variability across the complex. TWTT was converted to sediment thickness using two sound velocity choices—1500 and 1750 m s⁻¹—in order to bracket a likely value based on reported measured velocities through marine tills[62,63]. Sediment volumes (Supplementary Table 1) were calculated based on the mean thickness of 2500 CHIRP-derived data points for each sound velocity choice, and the area of a manually mapped outline of the GZWs interpreted from the multibeam surfaces. Grounding line width was measured from the mapped multibeam surface.

The local palaeo-sediment flux is unknown and reliable analogue fluxes are hampered by poor constraints on the mode(s) of subglacial sediment transport, particularly the thickness of a mobile till layer[64–68] and the volume of sediment transported in basal ice[69,70]. Despite the assumptions required to approximate these variables, and genuine spatial and temporal contrasts in both transport mode/rate and sediment supply, flux estimates reported from a variety of research approaches converge around 1–2 orders of magnitude. Here our approach encapsulates the range of uncertainties revealed by previous studies yet without specifically quantifying each source of error, in order to place upper and lower bounds on the possible duration of the readvance event.

Using subglacial sediment fluxes ranging from 50 to 8000 m³ a⁻¹ m⁻¹ (refs. [65,71]) we calculate a range of durations for the build-up of the GZW complex following Eq. 1 (Supplementary Table 1, Supplementary Fig. 2). Estimates from previous studies converge around 100–1000 m³ a⁻¹ m⁻¹ (refs. [32,66–70,72]), bracketing the

formation time for GZW1 as c. 80–1000 years, and for the whole grounding zone complex as c. 100–1600 years. This is, to an order of magnitude, consistent with using a catchment-wide sediment yield from ref. [32], although these modelled potential yields should be used as indicative only, rather than at face value.

To estimate the range of possible ice volumes discharged during the interpreted southern JOIDES readvance event (Supplementary Table 2), we consider event durations of 100, 600 or 1600 years, and flow velocities from 200–800 m a⁻¹, based on highly elongate glacial lineations that we interpret as a signature of fast flow. We thereby derive an along-flowline distance that would be discharged through the grounding line in each case, and using the glacial lineation distribution, we outline a corresponding zone of the ice sheet behind the readvance grounding line position that was likely discharged during the event (Fig. 4g). Within this zone, we sum the ice thickness values per grid cell area, derived from a hypothetical ice surface generated by projecting a simple parabolic profile, $h(x) = c.x^{1/2}$ (ref. [73]), inland from the mapped grounding line. We use values for constant c from 2.0–3.0 (ref. [74]) and satisfy a condition of grounded ice in all grid cells within the defined catchment. The parabola with constant $c = 2.5$ gives a comparable total ice volume to output from the ice sheet model described in refs. [7,75], for a timeslice whose ice margin and surface profile most closely match the configuration indicated by geomorphological evidence. For flowline lengths > 500 km (e.g., duration 1600 years and high velocity) the catchment definition becomes highly speculative. Scaling ice volumes similarly to our other scenarios gives a result well within the range of annual discharges (Gt a⁻¹) for better constrained scenarios, though these extreme case values should not be considered robust.

**Data availability.** Raw acoustic data from NBP1502A and legacy data displayed in Figs. 2–5 are available from www.marine-geo.org and from http://oden.geo.su.se. Integrated multibeam grids are available from the corresponding author upon request.

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

## Acknowledgements

We are grateful to the crew of the NBP and ASC support staff for a successful expedition. This work was supported by the National Science Foundation (NSF-PLR 1246353 to J.B. A.) and the Swedish Research Council (D0567301 to S.L.G.).

## Author contributions

S.L.G., L.M.S., A.R.W.H. and L.O.P. collected and processed data on NBP1502A. S.L.G. reprocessed the legacy data, performed the geomorphological mapping, acoustic analyses and readvance calculations. S.L.G., L.M.S., A.R.W.H., L.O.P. and J.B.A. contributed to data interpretations. S.L.G. wrote the manuscript, with contributions from all authors.

## Additional information

**Competing interests:** The authors declare no competing interests.

