## [Peer Review File · Nature Communications]

Reviewers' comments:

Reviewer #1 (Remarks to the Author):

The paper uses multibeam bathymetry to interpret a reconfiguration of ice flow in South Victoria Land in response to regional grounded ice sheet retreat in the Ross Sea. I focus my major comments on the three key conclusions cited at the end of the introduction. While I personally enjoyed the manuscript, I felt this was more for specialist interests, and that there are fundamental flaws in the assumptions used to link these data being related to a Holocene sea level.

i) reversal of ice flow direction in southern Drygalski Trough

This is a topic that was covered earlier by Greenwood et al., (2012), but has since been revised by Halberstadt et al., 2016 and Lee et al., 2017 who demonstrated some issues with that interpretation. I agree this is a complicated region and there was some reorganization of flow in this region, and a complete flow reversal probably did occur in the northernmost part of the Southern Drygalski Trough.

Although this is a key conclusion, there is not a huge amount of detail on these key features in this location, and only a sketch map is provided as evidence of this (Fig 3A). The supplementary figure 1, I received is not high enough resolution to see these features in detail, and I base my comments on the data set used in Lee et al., 2017 (Geology). I suggest the incorporation of the Lee dataset is also required here for this to be more complete, as aspects of the interpretation rely on these data (these data are available at <http://get.iedadata.org/doi/323884>).

Lee et al., show bifurcation of flow out of Mawson Glacier feeding into the Southern Drygalski Trough, this includes flow to the north and south (their figure 2A and 2B). These data show the lineations shown in Fig 3E backstep directly into the Mawson Glacier, so are not the continuous flowset as presented in Fig 3B – which is provided as the main evidence for this reversal (lines 137-139).

However, because of this above point, I do agree they are unlikely to be contemporaneous (as noted on lines 141), and represent a localized shift in flowline orientations. In the northern part of Southern Drygalski Trough, Lee et al., also map these lineations. However, they also note many transverse ridges overlaying these lineations, which are very clear but not mapped in this study. Lee et al., in their section on the "Transverse Ridges" noted the ambiguity in this region, and proposed the ice sheet reached flotation point earlier in the deeper regions of this trough. This could allow for some reorganisation of flow as proposed in this new manuscript. The lineations in Figure 3E appear to be much "fresher" in appearance, and could represent a later phase of advance than those in the deeper basin. Further to the north in this trough, the lineations are far less discrete, and have a drumlin type appearance, overprinted by transverse ridges. While this supports the conclusion presented in this manuscript and differs slightly from that presented in Lee et al., 2016, I am not convinced the implications of the event are as significant or as novel as presented in this manuscript.

ii) a retreat and readvance event in southern JOIDES Trough linked directly to Southern Victoria Land catchments, with a spatial scale >50 km, duration of 10s-100s years and sea level contribution equivalent to today's Antarctic discharge;

I have major concerns about the method used to arrive at this conclusion. There is no evidence from the data provided that there was a wholesale retreat and then major readvance that contributed significantly to sea level change in the Holocene. While there may be some flow relaxation and advance associated with deglaciation of the deep troughs and an associated decrease in buttressing from adjacent ice masses, this study does not constrain the extent of this

relative to sea level.

The assumptions involved for calculation of the duration of this advance are not valid. This requires a reliable estimate of subglacial sediment fluxes, and none of the studies cited provide this. A range of 200-1000 m³ a⁻¹ m⁻¹ is given based on references 62-65, and it is cited these are observed fluxes, which they are not. I very strongly believe the numbers produced in this section are not defensible, and should not be published.

- Ref. 62 is multibeam data only, and relies on subjective assumptions and thus to cite another multibeam estimate for this flux rate is circular.
- Ref 63 is a geophysical study from the modern grounding line. To calculate this sediment flux they assume the grounding line has been stable for a millennium, but the geological evidence for this is suspect and may have been much longer (closer to 5ka). They also cite “on the order of” 10^{>2} m³ a⁻¹ m⁻¹ which is less than the estimate used in the Greenwood et al., calculations.
- Ref 64 is from the North Sea, so is not really valid for this setting, but if used as an upper estimate is 8000 m³ a⁻¹ m⁻¹.
- Ref 65 rightly point out the additional importance of englacial debris in calculating basal debris fluxes (which could add significantly more sediment to GZW formation than previously considered). It also highlighted the importance of calculating melt rates accurately, and this influences whether this sediment would be deposited in a grounding zone or more distally. To use this study from a single location to calculate past GZW formation rates is premature.

Even without the consideration of englacial debris complicating the estimates, the uncertainties associated with delivery of subglacial are too great to be reliable. These rely on accurate estimates of past ice velocities and more critically deforming till depths - estimates of which can range from a few cm to several metres (a large debate its own right). These are fundamental for calculating this flux and the uncertainties for both of these pushes out the range of sediment fluxes far greater than cited here (in both directions) and therefore the link to sea level is fundamentally flawed.

With our current state of knowledge, the numbers produced are meaningless – particularly as it is claiming to identify sea level contributions to a resolution of 0.1 - 0.24 mm/a. Melt rates, accommodation space and underlying geology would also play a critical role on GZW volumes, and these are also poorly constrained. This distracts from the presentation of what is otherwise a nice dataset and robust interpretations. I also note these numbers and methods are presented in the accompanying manuscript (and thus are not novel).

It also does not really constrain the extent of the retreat prior to this “readvance” it has little context for Holocene sea level budget, other than the ~50 km shown in figure 4C (note, there is no scale shown in figure 4C?). However, this is a relatively small readvance - and was probably associated with ice loss elsewhere, so in effect is a net sea level drop? I agree this could have been associated with deflation of the ice stream profile as this ice accelerated following a loss of buttressing from ice in the central embayment (as noted on line 314-317), but the link to a significant sea level contribution is tenuous. I found link to the chronological constraints also needs better development of the argument, and have some comments/thoughts on this below, that may help further with this discussion.

-I agree the 8.6 ka age at Coulman High probably represents retreat of ice sourced from the south of Ross Island, which represented an initial loss of an ice shelf and regrowth. However, this Coulman High site today experiences periodic regrowth of an ice shelf (but sediment is winnowed and thus the recent history in that core is condensed). An alternative interpretation is that this initial ice shelf retreat was preserved as it was buried by an active sediment supply derived from a nearby grounding line (i.e the calving line and grounding line were in close proximity during

retreat) rather than widespread ice shelf readvance – is there an evidence from the multibeam of any such process? – e.g. basal crevasse squeeze ridges. I do agree the local GZW readvance probably occurred after this (albeit this is a minimum constraint for the retreat age and could be older)

- Please explain better your links to the cosmogenic data from Mackay Glacier, and coastal open marine conditions as evidence for this readvance (as noted on lines 256-259). As written, all these state is that final retreat was complete by this time – what is the support for the readvance?

iii) grounding line oscillations or reconfiguration of a remnant Crary Bank ice rise.

As with the Southern Drygalski, this region is interesting and very complex. I agree there have to multiple phases of ice flow directions. However, determining a relative timing of events is difficult. The red moraines in Figure 5C do not display any overlapping behavior to determine this. An alternate solution is that the red moraines have been formed during the advance and grounding of ice? - i.e. the Crary Bank may have acted to have buttress ice shelves, resulting in over-thickening and eventual grounding in the deep troughs? As these moraine very substantial features, its seem hard to reconcile with ice divide flow that would be predicted as the bank became isolated from grounded ice in the adjacent trough - but I concede this is a subjective interpretation (but so is the one presented in the manuscript).

I agree with the authors, there is enough evidence here to cite Crary Bank ice rise was the final area to decay in this local region (based on the green moraine overlapping the white lineations in Figures 5C and D) – but maybe not within the Ross Sea as whole due to a lack of chronological constraints (e.g. residual coastal ice). This is also conclusion also noted in Halberstadt et al., 2016.

Are they novel and will they be of interest to others in the community and the wider field?

This is the second paper from this authorship group (following Halberstadt et al., 2016; Cyroshpere) on this topic recently and utilizing the same dataset. The idea of evolving flowlines through a glacial cycle is not new, with the Halberstadt et al., 2016 paper proposing a dynamic flow-switching model. The supporting manuscript (Simkins et al., 2017) also presents a case for GZW duration using the same method as in this paper. I am unconvinced how significant these additional observations are for the “big picture” implications relating to sea level in this paper. While it is a spectacular dataset, and most of the interpretations are sound, major refocusing of the implications are needed to highlight if these new data and observations for it to be deemed novel.

If the conclusions are not original, it would be helpful if you could provide relevant references. Is the work convincing, and if not, what further evidence would be required to strengthen the conclusions?

The method to calculation the duration of GZW still-stand component is not valid as modern processes are not well-enough constrained to take this approach. The level of precision (to the sub-mm scale of sea level rise per year) is claimed that is misleading. The supporting paper also uses this methodology, but is perhaps less important to their interpretations.

On a more subjective note, do you feel that the paper will influence thinking in the field?

I personally found the paper interesting, but probably of special interest. I encourage the dataset to be published in some form, but I am unconvinced it will influence larger thinking in the field in its current form.

Please feel free to raise any further questions and concerns about the paper.

As this is the most to up to data "synthesis", then it probably should incorporate the dataset published by Lee et al., 2017, as this has major implications for the ice flow interpretation in the southern Drygalski Trough. These data are freely available as noted above.

Rob McKay

Reviewer #3 (Remarks to the Author):

Greenwood et al. provide an interesting study with fantastic new swath-bathymetric imagery showing close interaction between general post-Last Glacial Maximum retreat across the western Ross Sea Embayment (RSE) shelf and corresponding subsequent predominance and re-advance through glacial outlets in the Trans Antarctic Mountains onto the RSE shelf. The study reveals that the western RSE seemed to have deglaciated in a much different fashion, likely due to the close proximity of the East Antarctic Ice Sheet (EAIS) and the independent behavior of EAIS-draining Trans Antarctic Mountain (TAM) glaciers once the grounding line retreated from the RSE shelf. This finding importantly sheds light on the complex and ambiguous retreat pattern recorded previously by glacial landforms on the western RSE shelf.

However, I see some limitations to their conclusions, since chronologic constraints based on ages from marine sediment cores are rather weak, entirely base on a single age reported in McKay et al. (2016, GEOLOGY), and are used to infer ice-sheet retreat in a relatively large area.

Furthermore, the calculated formation periods for their imaged grounding-zone wedges are given in a very broad range (80-500 yrs and 100-800 yrs, respectively), relying on grounding-line sediment fluxes taken from other drainage basins. This generally raises concerns because these fluxes may vary significantly between different drainage basins, and may be entirely different for glaciers draining the Southern Victoria Land. I acknowledge that their approach at this time seems to be the best way forward, however, I would suggest that the authors at least mention the urgency for recovering more marine sediment core material during future campaigns that may provide much better limitations for grounding-line retreat from the respective wedges, further allowing much more precise constraints on formation periods of the single wedges and the entire complex. The authors should make this clear and also say that this may well lead to reviewing their whole story. Because right now their main conclusion reads very definite (e.g. last sentence of the abstract "...drove an ice sheet readvance of sufficient magnitude to influence Holocene sea level"), which may change after getting better in-situ ages. This really has to be made clear since right now it remains rather vague, at what time the re-advance took place and how long it may have lasted, since the present study entirely relies on geophysical data. Again - I generally suggest more careful wording for these issues and urge for including a few sentences that point out the importance of more reliable age control - i.e. taking sediment cores from the GZWs - for both the overall time frame of events, and GZW formation periods / calculation of local sediment fluxes. Besides that, the manuscript is very well written, making it easy and enjoyable to read. I like their story and would like to see it published after addressing my more general comments above and a few minor comments that I list below.

Minor comments:

Line 36: Capitalize "Last Glacial Maximum".

Line 55: Avoid questions in the manuscript.

Line 62: Write "decades to centuries" instead of "10s-100s".

Line 85: Replace "seascape" with "seafloor".

Line 121: This only applies if moraines were formed at the same time. Is there core evidence (ages)? If not, phrase more carefully here.

Line 161: Write "oriented perpendicular".

Line 174: Write "ENE".

Line 182: Replace "ever" with "increasingly".

Line 195: Write "the most distal wedge".

Line 270: What is the Ross Sea ice sheet? Do you mean formerly grounded ice in the RSE? Replace "non-monotonic" with "heterogenic".

Lines 288-301: Use the term "stillstand".

Lines 319-321: Very nice and interesting result and nicely comparable to recent events at Larsen C (maybe cite).

Figures:

Fig. 1: Give more regional context for Antarctic overview map, e.g. "WAIS", "EAIS", Amundsen Sea, Weddell Sea, ... just for reference. And maybe include general coordinates. Maybe adjust color range in main figure so that troughs in RSE become more pronounced and give color code in legend. Explain "T.A.M." in caption.

Fig. 2: Label "Drygalski Trough" in A. Again: also give more pronounced color range and give color code in legend. Maybe also give a legend with explained mapped landforms. F: I honestly cannot clearly see the crevasse-squeeze ridges. Give a zoom-in.

Fig. 3: Referring to this figure: Mention in text that further mapping just south of 76°S and just E and W of 164°E could be crucial in order to explain different MSGL orientations. It would be good to see the actual locations of sub-bottom profiles in C and D - maybe create a zoom-in of A as an inset.

Fig. 4: It would be good to see the locations of sub-bottom profiles on bathymetric maps and not only on the sediment thickness map.

Fig. 7: Give direction of reconstructed flow lines and use darker color.

Thank you.

J. P. Klages

We thank both reviewers for their thorough appraisal of our manuscript, and for pointing out aspects of our work that were in need of further attention. We respond here to their various comments. All references to line numbers in our replies refer to the new version of the document, unless otherwise stated.

Reviewer #1, Rob McKay

The paper uses multibeam bathymetry to interpret a reconfiguration of ice flow in South Victoria Land in response to regional grounded ice sheet retreat in the Ross Sea. I focus my major comments on the three key conclusions cited at the end of the introduction. While I personally enjoyed the manuscript, I felt this was more for specialist interests, and that there are fundamental flaws in the assumptions used to link these data being related to a Holocene sea level.

i) reversal of ice flow direction in southern Drygalski Trough

This is a topic that was covered earlier by Greenwood et al., (2012), but has since been revised by Halberstadt et al., 2016 and Lee et al., 2017 who demonstrated some issues with that interpretation. I agree this is a complicated region and there was some reorganization of flow in this region, and a complete flow reversal probably did occur in the northernmost part of the Southern Drygalski Trough.

Although this is a key conclusion, there is not a huge amount of detail on these key features in this location, and only a sketch map is provided as evidence of this (Fig 3A). The supplementary figure 1, I received is not high enough resolution to see these features in detail, and I base my comments on the data set used in Lee et al., 2017 (Geology).

-> A large swarm of lineations in southern Drygalski are clearly linked with southward flow from (at least) Mawson & Mackay outlets, and possibly others on the SVL coast. A small group of lineations around Drygalski Ice Tongue are oriented in the opposing direction, incompatible with simultaneous southward flow. These are few, small-scale and subtle in their relief, and therefore there is limited 'amount of detail' to present. That there are only a few features does not negate their significance, they still falsify a hypothesis of southward-only flow: these are incompatible with the rest of the population and must therefore argue for a flow reversal. We illustrate these features in Fig 3B, and argue for their incompatibility with southward flow based on the wider distribution of features in Fig 3A (we note that individual features are mapped true to scale and occurrence in Fig 3A, not as a sketch map). In our revised submission, we have increased the resolution of the supplementary figure, so that small-scale features are resolvable when zoomed-in on-screen.

I suggest the incorporation of the Lee dataset is also required here for this to be more complete, as aspects of the interpretation rely on these data (these data are available at <http://get.iedadata.org/doi/323884>).

-> our interpretations and conclusions are entirely independent of the Lee dataset (referring to the new data collected by Lee et al from RV Araon, available from the above link), since that was unavailable at the time of our mapping work. The archived (NBP) data that they compile is the same as that we have independently compiled and re-processed (cleaned) here, and which we have interpreted at the full on-screen resolution possibly achieved by that data.

Notwithstanding this, we have taken the opportunity to incorporate the RV Araon data and conduct full landform mapping from it. We find that our main conclusions do not change and we comment on specific aspects in reply to the following two points raised by the reviewer.

Lee et al., show bifurcation of flow out of Mawson Glacier feeding into the Southern Drygalski Trough, this includes flow to the north and south (their figure 2A and 2B). These data show the lineations shown in Fig 3E backstep directly into the Mawson Glacier, so are not the continuous flowset as presented in Fig 3B – which is provided as the main evidence for this reversal (lines 137-139).

-> We agree that many of the southern Drygalski lineations mapped in Fig 3A and visualised in 3E withdraw into Mawson Glacier. We discuss this in the original text (new lines 123-125, 151-152, 238-240) and we have added clarification into our revised 1st paragraph of the section 'Flow reversal in southern Drygalski trough'. These lineations notwithstanding, there are also lineations that are not oriented with flow out of Mawson Glacier, and instead are aligned across the glacier outlet (north-south), in the central axis of Drygalski trough immediately in front of Mawson outlet. New mapping from the Araon data further supports this (see updated Figs 2A and 3A). We argue that the position and orientation of these lineations precludes Mawson Glacier as a supply route, either to the north or to the south, and provide a false impression of a continuous flowset which we argue must in fact be a combination of two opposing, non-contemporaneous flows.

However, because of this above point, I do agree they are unlikely to be contemporaneous (as noted on lines 141), and represent a localized shift in flowline orientations. In the northern part of Southern Drygalski Trough, Lee et al., also map these lineations. However, they also note many transverse ridges overlaying these lineations, which are very clear but not mapped in this study. Lee et al., in their section on the "Transverse Ridges" noted the ambiguity in this region, and proposed the ice sheet reached flotation point earlier in the deeper regions of this trough. This could allow for some reorganisation of flow as proposed in this new manuscript. The lineations in Figure 3E appear to be much "fresher" in appearance, and could represent a later phase of advance than those in the deeper basin. Further to the north in this trough, the lineations are far less discrete, and have a drumlin type appearance, overprinted by transverse ridges.

-> We had originally mapped clusters of recessional moraines along Drygalski Trough from our NBP data compilation; we have now mapped the additional landforms revealed by the extended data coverage from the Araon data collected by Lee et al. We find that some of these 'transverse ridges' that flank Cray Bank are associated with small-scale fluting on top (extension of the assemblage shown in Fig 2C & 5D); others overprint longer lineations that correspond to an earlier flow event with a more distal ice margin. None of these newly mapped features contradicts our earlier interpretation of non-contemporaneous northward and southward flow through southern Drygalski Trough (i.e. a flow reversal), in fact we would argue they support this conclusion.

While this supports the conclusion presented in this manuscript and differs slightly from that presented in Lee et al., 2016, I am not convinced the implications of the event are as significant or as novel as presented in this manuscript.

-> We suggest that the significance of our observations in southern Drygalski Trough is not merely in the finding of one flow reconfiguration (although this is interesting), but rather that this is just one of several reorganisations of ice flow in the western Ross Sea that we present in this paper, and that the flow switch in southern Drygalski Trough is, we find, associated with a significant grounding line readvance in this SVL sector of the EAIS.

ii) a retreat and readvance event in southern JOIDES Trough linked directly to Southern Victoria Land catchments, with a spatial scale >50 km, duration of 10s-100s years and sea level contribution equivalent to today's Antarctic discharge;

I have major concerns about the method used to arrive at this conclusion. There is no evidence from the data provided that there was a wholesale retreat and then major readvance that contributed significantly to sea level change in the Holocene. While there may be some flow relaxation and advance associated with deglaciation of the deep troughs and an associated decrease in buttressing from adjacent ice masses, this study does not constrain the extent of this relative to sea level.

-> In this part of the paper, we report grounding line landforms that unquestionably require grounding line retreat followed by grounding line readvance. Our evidence requires that the spatial scale of this grounding line event is at least 50 km (lines 185-187). We also identify additional, but equivocal, evidence that the spatial scale may have been greater, up to 150 km; we discuss why these data are less secure than those constraining the minimum 50 km advance (lines 187-200). It is not clear (not quantified) what the reviewer means by "wholesale retreat and major readvance", nor what would distinguish this description from "flow relaxation and advance". Ambiguity in the way that grounding line movements are described, and the (subjective?) significance attached to them, is precisely our motivation for attempting to quantify the scale (spatial) and the duration (temporal) of the event that we observe, and for then exploring possible drivers behind an event of the quantified magnitude.

It was not our intention to quantify this readvance event's specific contribution to the Holocene sea level budget (we comment further on this below). We rather wish to provide some context for the magnitude of the event that we have reconstructed, by comparison with contemporary Antarctic ice sheet discharge. We have paid careful attention to our wording of such comparisons in our revised manuscript, and replaced our earlier sea level comparison with a Gt-discharge comparison to present-day systems.

The assumptions involved for calculation of the duration of this advance are not valid. This requires a reliable estimate of subglacial sediment fluxes, and none of the studies cited provide this. A range of 200-1000 m³ a⁻¹ m⁻¹ is given based on references 62-65, and it is cited these are observed fluxes, which they are not. I very strongly believe the numbers produced in this section are not defensible, and should not be published.

- Ref. 62 is multibeam data only, and relies on subjective assumptions and thus to cite another multibeam estimate for this flux rate is circular.
- Ref 63 is a geophysical study from the modern grounding line. To calculate this sediment flux they assume the grounding line has been stable for a millennium, but the geological evidence for this is suspect and may have been much longer (closer to 5ka). They also cite "on the order of" 10² m³ a⁻¹ m⁻¹ which is less than the estimate used in the Greenwood et al., calculations.
- Ref 64 is from the North Sea, so is not really valid for this setting, but if used as an upper estimate is 8000 m³ a⁻¹ m⁻¹.
- Ref 65 rightly point out the additional importance of englacial debris in calculating basal debris fluxes (which could add significantly more sediment to GZW formation than previously considered). It also highlighted the importance of calculating melt rates accurately, and this influences whether this sediment would be deposited in a grounding zone or more distally. To use this study from a single location to calculate past GZW formation rates is premature.

Even without the consideration of englacial debris complicating the estimates, the uncertainties

associated with delivery of subglacial are too great to be reliable. These rely on accurate estimates of past ice velocities and more critically deforming till depths - estimates of which can range from a few cm to several metres (a large debate its own right). These are fundamental for calculating this flux and the uncertainties for both of these pushes out the range of sediment fluxes far greater than cited here (in both directions) and therefore the link to sea level is fundamentally flawed.

-> *We agree with both reviewers that constraining the timing of an event via the sediment flux approach we have used is less than ideal – we would prefer to be able to do so with a robust absolute (e.g. radiocarbon) chronology. However, despite hundreds of cores having been taken from the western Ross Sea over the last years-decades (including our own recent cruise that specifically targeted the grounding line landforms with which this paper is concerned), a reliable radiocarbon chronology for deglaciation, with small enough errors to constrain short-term grounding line change, is unfortunately still lacking, for reasons well-rehearsed by several authors (e.g. Andrews et al 1999, Quaternary Research; Anderson et al 2014, QSR; McKay et al 2016, Geology). In the absence of a good absolute chronology, we follow several other research groups (e.g. Graham et al 2010, JGR; Jakobsson et al 2012, QSR; Livingstone et al 2016, J Glaciology; Bart et al 2017, Scientific Reports) in taking a 'next best' approach.*

We also acknowledge that estimates of sediment fluxes are difficult to constrain. It is precisely for this reason that we explore a range of sediment flux values, in order to place upper and lower bounds on the duration of our event. In doing so, we capture the uncertainty in present understanding of sediment fluxes, without needing to quantify or better constrain each underlying assumption that accounts for that uncertainty. We simply seek to bracket possibilities. We argue that constraining maximum or minimum possible rates of grounding line change and grounding line stability is a worthwhile endeavour, irrespective of the exact age of the event.

However, we also recognise that our motivations and our approach could have been better described, the uncertainties that the reviewer highlights should have been acknowledged and discussed, and our treatment of the literature on sediment fluxes should have been more rigorous and explicit in order for our results to be credible. We have therefore revised our Methods section to incorporate a more detailed presentation of and motivation for our approach and, indeed, reviewed the range of sediment fluxes we adopted in our calculations of max/min event duration. In conjunction with this we have added Supplementary figure 2, which models the flux-duration relationship across a wide range of values, and highlights an envelope within which many literature estimates converge. We additionally consider a second, semi-independent method for estimating GZW formation time, following Bart et al 2017 (Sci. Reports) and recommended by Golledge, Levy, McKay et al. 2013, QSR. With these revisions, we believe our conclusions are more robust and we thank both reviewers for drawing our attention to an aspect of the manuscript that needed improvement.

In specific reference to the flux values used, and which the reviewer comments on above (using the original reference numbers):

- Ref. 62 indeed should not have been used. That it is a geophysical study is not of concern, but rather that it doesn't directly use the geophysical data to derive a flux; their flux value is hypothetical, based also on other literature and assumptions (as we do here). Citing this article was an oversight that we have corrected, arising from a value used that is consistent with much other sediment flux literature.

- Ref. 63 describe a GZW volume 'on the order of' $10^5 \text{ m}^3\text{m}^{-1}$, a duration 'probably stable for a millennium' and a sediment flux therefore 'on the order of' $10^2 \text{ m}^3\text{a}^{-1}\text{m}^{-1}$. The implied flux may therefore take values from $100\text{-}999 \text{ m}^3\text{a}^{-1}\text{m}^{-1}$ and a duration of 5000 years or 1000 years (both order 10^3) still yields a flux of the same order as cited. The authors comment on consistency with a flux of $150 \text{ m}^3\text{a}^{-1}\text{m}^{-1}$ based on findings reported by Kamb 2001 and Engelhardt & Kamb 1998. Bart et al 2017 cite Ref. 63's flux as $200 \text{ m}^3\text{a}^{-1}\text{m}^{-1}$ based on a volume of $2 \times 10^5 \text{ m}^3\text{m}^{-1}$ corresponding to the imaged and

non-imaged part of the contemporary GZW. In our revisions, we consider the widest range implied by Ref. 63's result, i.e. 100-1000 m³a⁻¹m⁻¹.

- Ref. 64 is a geophysical study with a well-constrained independent chronology. That it comes from a different ice sheet setting does not, we argue, diminish its applicability here; there is no a priori reason why subglacial transport of sediments should (not) differ with geography. In Supplm Fig 2 we extend our calculations up to and including the 8000 m³a⁻¹m⁻¹ flux inferred by these authors.

- Ref. 65 quantifies basal ice debris content, translated into likely fluxes over time. The order of magnitude is comparable to that of sediment transport through a mobile subglacial till layer and a total flux should include both. Ref. 65 envisages that basal debris entrainment is favoured during ice stream stagnation (loss of frictional heat) and that it should be low during fast flow episodes (basal melt prevents significant entrainment). Modes of sediment transfer may therefore cycle between subglacial and englacial. Nonetheless, this is a source of uncertainty in grounding line depositional flux that would tip our duration calculations towards either the lower (higher flux) or upper (slower flux) bound.

We base our revised calculations on literature encompassing direct measurement (e.g. Engelhardt & Kamb 1998; Kamb 2001), sediment geotechnical properties (e.g. Ó Cofaigh et al 2007), contemporary and palaeo geophysical inversion (e.g. Anandakrishnan et al 2007; Nygård et al 2007; Hooke & Elverhøi 1996; Elverhøi et al 1998; Livingstone et al 2016) and quasi-physical modelling (e.g. Hooke & Elverhøi 1996; Bougamont & Tulaczyk 2003; Golledge et al 2013). We note that these different research approaches, notwithstanding the assumptions that they employ and the uncertainties implicit in each, typically converge on fluxes from about 100-1000 m³a⁻¹m⁻¹.

Furthermore, we explore a second, semi-independent approach adopted by Bart & Owalana (2012, QSR), Bart et al 2017 (Scientific Reports) and recommended by Golledge et al 2013 (QSR: "we suggest that these simulations might provide a useful addition to the geological sources of information employed in quantifying sedimentary landform generation rates (eg grounding zone wedges) from which ice sheet or ice stream retreat rates may be inferred"), that uses basal erosion rates to generate a catchment yield, rather than using transport rates. We extract the net erosion rates modelled for our catchment by Golledge et al 2013 from their retreat event simulation and generate a catchment yield. Assuming that net yield translates to an annual grounding line flux (paying no regard to mode of transport), then our results from this approach fit well within the range of durations that our original, analogue flux approach provides.

Neither approach is perfect, both relying on a range of assumptions, but since here we intend to bracket a range of possibilities, we argue that our results are reliable in the context of this purpose, and our revised text clarifies this purpose.

With our current state of knowledge, the numbers produced are meaningless – particularly as it is claiming to identify sea level contributions to a resolution of 0.1 - 0.24 mm/a. Melt rates, accommodation space and underlying geology would also play a critical role on GZW volumes, and these are also poorly constrained. This distracts from the presentation of what is otherwise a nice dataset and robust interpretations. I also note these numbers and methods are presented in the accompanying manuscript (and thus are not novel). It also does not really constrain the extent of the retreat prior to this "readvance" it has little context for Holocene sea level budget, other than the ~50 km shown in figure 4C (note, there is no scale shown in figure 4C?).

-> Unfortunately we can only constrain a minimum distance, as discussed above. The evidence for spatial magnitude is clearly discussed in lines 185-202. Thanks for drawing our attention to a missing scale bar – we have added this to Fig 4C.

However, this is a relatively small readvance - and was probably associated with ice loss elsewhere, so in effect is a net sea level drop? I agree this could have been associated with deflation of the ice stream profile as this ice accelerated following a loss of buttressing from ice in the central embayment (as noted on line 314-317), but the link to a significant sea level contribution is tenuous.

-> We would argue that 50 km (possibly much larger) is not an insignificant readvance magnitude, which is why we present a comparison with discharge from contemporary outlet systems. Even without comparable discharge volumes and sea level implications, the readvance that we observe here is linked with significant alteration of ice flow structure and ice sheet profile.

Our intention was not to make a specific claim regarding the Holocene sea level budget, significant or otherwise. Our calculation is neither a net balance result (purely a quantification of discharge) nor does it account for the palaeo land-ocean geography and basin volume. We estimate an order-of-magnitude discharge from the readvance event and contextualise our result (is it a 'relatively small advance?') by comparison with modern ice sheet discharge (expressed in our original version as sea level equivalent). I agree that quoting SLE values as fractions of a millimetre gives a false impression of precision; this is a consequence of the scaling between Gt discharge and global SLE. In our revised manuscript, we instead use Gt-discharge comparisons with present-day catchments and we emphasise that our comparisons are for context only; we have made careful changes to our wording to avoid any implication of a discharge or sea level claim specific to the Holocene sea level budget. I also note that we have revised the discharge calculations using event durations of 200 and 600 years, instead of 100 and 500 years in the original version. This takes into account revised duration calculations, as above.

I found link to the chronological constraints also needs better development of the argument, and have some comments/thoughts on this below, that may help further with this discussion.

- I agree the 8.6 ka age at Coulman High probably represents retreat of ice sourced from the south of Ross Island, which represented an initial loss of an ice shelf and regrowth. However, this Coulman High site today experiences periodic regrowth of an ice shelf (but sediment is winnowed and thus the recent history in that core is condensed). An alternative interpretation is that this initial ice shelf retreat was preserved as it was buried by an active sediment supply derived from a nearby grounding line (i.e. the calving line and grounding line were in close proximity during retreat) rather than widespread ice shelf readvance – is there any evidence from the multibeam of any such process? – e.g. basal crevasse squeeze ridges. I do agree the local GZW readvance probably occurred after this (albeit this is a minimum constraint for the retreat age and could be older)
- Please explain better your links to the cosmogenic data from Mackay Glacier, and coastal open marine conditions as evidence for this readvance (as noted on lines 256-259). As written, all these state is that final retreat was complete by this time – what is the support for the readvance?

-> I am not sure I fully understand the distinction the reviewer makes in his first point, above. That we cannot discern the magnitude of ice shelf re-growth? We agree: this is implicit in our two possible scenarios concerning the CH date and our observed readvance, and we have further clarified this distinction (lines 284-290).

The terrestrial dates are indeed for final deglaciation; they do not offer evidence for the readvance itself. They must however, by definition, act as a minimum age for the readvance event, and thus in conjunction with the Coulman High date constrain the whole deglacial sequence (first retreat; readvance; final retreat) to a period of ~2000 years. We have clarified this in our revised text.

iii) grounding line oscillations or reconfiguration of a remnant Crary Bank ice rise.

As with the Southern Drygalski, this region is interesting and very complex. I agree there have to be multiple phases of ice flow directions. However, determining a relative timing of events is difficult. The red moraines in Figure 5C do not display any overlapping behavior to determine this.

-> We agree, our solution to the sequence of Crary Bank's demise might not be unique – it is one hypothesis. We have acknowledged the difficulty in determining even a relative chronology in our revised text. We do, however, suggest that our solution is the simplest explanation of the data.

An alternate solution is that the red moraines have been formed during the advance and grounding of ice? - i.e. the Crary Bank may have acted to have buttress ice shelves, resulting in over-thickening and eventual grounding in the deep troughs? As these moraine very substantial features, it seems hard to reconcile with ice divide flow that would be predicted as the bank became isolated from grounded ice in the adjacent trough - but I concede this is a subjective interpretation (but so is the one presented in the manuscript).

-> Yes, this could also be a solution. It would require long-term cold-based ice on the bank top (not, in itself, a difficult interpretation) in order to preserve the moraines as advance features. However, we would also require preservation of small-scale landforms immediately behind the advancing grounding line (i.e. preserving the most recent moraine as the grounding line moves forward over it), while that advance regime is clearly capable of raising relief anew at each successive grounding line position. This is rather more awkward to accept. I am not sure what the reviewer implies by 'moraines [are?] very substantial features' – in coverage, certainly, there are many; but their amplitude is rather subtle. We note that similar along-contour moraines are reported along the submerged flank of Ross Island (Greenwood et al 2012, GSA Bulletin, their Fig. 3A), relating to its establishment (& shrinkage) as an independent ice body with radial flow, detached from grounded Ross Sea ice.

I agree with the authors, there is enough evidence here to cite Crary Bank ice rise was the final area to decay in this local region (based on the green moraine overlapping the white lineations in Figures 5C and D) – but maybe not within the Ross Sea as whole due to a lack of chronological constraints (e.g. residual coastal ice). This is also a conclusion also noted in Halberstadt et al., 2016.

-> We agree. Our original wording (original lines 279-80) was sloppy, and we have revised the text accordingly.

Are they novel and will they be of interest to others in the community and the wider field?

This is the second paper from this authorship group (following Halberstadt et al., 2016; Cyroshpere) on this topic recently and utilizing the same dataset. The idea of evolving flowlines through a glacial cycle is not new, with the Halberstadt et al., 2016 paper proposing a dynamic flow-switching model. The supporting manuscript (Simkins et al., 2017) also presents a case for GZW duration using the same method as in this paper. I am unconvinced how significant these additional observations are for the “big picture” implications relating to sea level in this paper. While it is a spectacular dataset, and most of the interpretations are sound, major refocusing of the implications are needed to highlight if these new data and observations for it to be deemed novel.

-> The idea of flow reconfiguration in an ice sheet is certainly not new, even in this sector of the Ross Sea. Greenwood et al 2012 found evidence for reorganisations, although new data of superior quality enable us (and others, eg Lee et al 2017) to re-assess some of those interpretations. Halberstadt et al

set out a framework of flow and retreat for the whole of the Ross Sea that recognises non-uniform behaviour (spatial and temporal). The Halberstadt et al paper does not, however, explicitly recognise any of the flow/grounding line reorganisations presented in this manuscript. Our three central findings – a grounding line readvance, its link to a flow reconfiguration in Southern Victoria Land, and reconfiguration(s) of a remnant Cray Bank ice rise – are new, not previously analysed or published, and illustrate the way that a relatively small source catchment may undergo major changes in response to regional grounding line change. These changes, we show here, may run counter to the regional direction of change (i.e. readvance, in response to regional retreat) and these changes may be sustained for centuries.

Our findings from the western Ross Sea are among a spate of recent, high profile papers that find reorganisations in different sectors of the Antarctic ice sheet following retreat from the LGM. These collectively raise questions of the vulnerability, stability or resilience of different sectors, and demand analysis of why different sectors may behave in different ways including, in cases such as we present here, advance for a period of centuries in the face of climate warming and sea level rise.

Our conceptualisation of the magnitude of the event that we reconstruct in terms of sea level was clearly misleading, and we have re-framed this aspect of the paper, making comparisons to contemporary discharge systems only in terms of volume contributions of ice, not a sea level equivalent. We also stress that our quantification of the magnitude of our observed readvance is intended only to set upper and lower bounds on a possible magnitude that should lie somewhere within this calculated envelope of possibilities.

If the conclusions are not original, it would be helpful if you could provide relevant references. Is the work convincing, and if not, what further evidence would be required to strengthen the conclusions?

The method to calculate the duration of GZW still-stand component is not valid as modern processes are not well-enough constrained to take this approach. The level of precision (to the sub-mm scale of sea level rise per year) is claimed that is misleading. The supporting paper also uses this methodology, but is perhaps less important to their interpretations.

*-> We have argued in our comments above that our general approach to calculating GZW formation time is valid, despite the uncertainties in present understanding of sediment fluxes. We recognise these uncertainties and, taking these into account, define an envelope of time within which our measured GZW complex can reasonably be expected to have formed. The resulting range of time (a few centuries) is comparable to expected error bars on even reliable radiocarbon dates from this setting and time period. We accept that our derivation of a possible sea level contribution may have been misleading; we attempted to provide a contemporary context for the magnitude of our event, not to claim a specific contribution to the Holocene sea level budget. The scaling relationship between volume discharge and global sea level equivalent also leads to a false impression of precision, since it is expressed as a fraction of a millimetre. **We have changed our 'context' variable to annual Gt discharge of ice, rather than converting to SLE.***

On a more subjective note, do you feel that the paper will influence thinking in the field?

I personally found the paper interesting, but probably of special interest. I encourage the dataset to be published in some form, but I am unconvinced it will influence larger thinking in the field in its current form.

Please feel free to raise any further questions and concerns about the paper.

As this is the most up to date "synthesis", then it probably should incorporate the dataset published by Lee et al., 2017, as this has major implications for the ice flow interpretation in the

southern Drygalski Trough. These data are freely available as noted above.

-> *We have noted above that our interpretations are entirely independent of the new data collected by Lee et al. Their data does, however, extend multibeam coverage in crucial areas of southern Drygalski Trough and is therefore an important complement to our work. Now that the dataset is publically available, we have conducted further landform mapping (individual feature mapping) from these data. The new mapping supports our earlier interpretations of flow reversal and ice rise (Crary Bank) reorganisation of the grounding line during stages of retreat.*

Reviewer #3, Johan Klages:

Greenwood et al. provide an interesting study with fantastic new swath-bathymetric imagery showing close interaction between general post-Last Glacial Maximum retreat across the western Ross Sea Embayment (RSE) shelf and corresponding subsequent predominance and re-advance through glacial outlets in the Trans Antarctic Mountains onto the RSE shelf. The study reveals that the western RSE seemed to have deglaciated in a much different fashion, likely due to the close proximity of the East Antarctic Ice Sheet (EAIS) and the independent behavior of EAIS-draining Trans Antarctic Mountain (TAM) glaciers once the grounding line retreated from the RSE shelf. This finding importantly sheds light on the complex and ambiguous retreat pattern recorded previously by glacial landforms on the western RSE shelf.

However, I see some limitations to their conclusions, since chronologic constraints based on ages from marine sediment cores are rather weak, entirely base on a single age reported in McKay et al. (2016, GEOLOGY), and are used to infer ice-sheet retreat in a relatively large area.

-> *A complete reconstruction of western Ross Sea deglaciation is indeed limited by a poor chronology, despite an abundance of new and archived cores from this region. Our own targeted coring of geomorphic features (as recommended below) on cruise NBP1502A was largely unsuccessful in recovering sufficient carbonate from appropriate stratigraphic positions to constrain the exact timing of grounding line change. We do not see that the lack of a robust chronology undermines the conclusions that we reach, which are based on configuration changes in the grounding line. Our ability to constrain rates of change is unfortunately limited to making estimates within upper and lower bounds. We therefore discuss max/min absolute ages for timing, using the McKay et al date from offshore combined with terrestrial ages for final retreat; **we have clarified how we use these sets of information in the revised text.** And we independently calculate upper/lower bounds estimates for the duration of a readvance event based on a range of possible sediment fluxes; see further response below, and to the first reviewer, above.*

Furthermore, the calculated formation periods for their imaged grounding-zone wedges are given in a very broad range (80-500 yrs and 100-800 yrs, respectively), relying on grounding-line sediment fluxes taken from other drainage basins. This generally raises concerns because these fluxes may vary significantly between different drainage basins, and may be entirely different for glaciers draining the Southern Victoria Land. I acknowledge that their approach at this time seems to be the best way forward, however, I would suggest that the authors at least mention the urgency for recovering more marine sediment core material during future campaigns that may provide much better limitations for grounding-line retreat from the respective wedges, further allowing much more precise constraints on formation periods of the single wedges and the entire complex. The authors should make this clear and also say that this may well lead to reviewing their whole

story. Because right now their main conclusion reads very definite (e.g. last sentence of the abstract "...drove an ice sheet readvance of sufficient magnitude to influence Holocene sea level"), which may change after getting better in-situ ages. This really has to be made clear since right now it remains rather vague, at what time the re-advance took place and how long it may have lasted, since the present study entirely relies on geophysical data. Again - I generally suggest more careful wording for these issues and urge for including a few sentences that point out the importance of more reliable age control - i.e. taking sediment cores from the GZWs - for both the overall time frame of events, and GZW formation periods / calculation of local sediment fluxes.

-> *As we've commented above in response to the first reviewer, we agree that an absolute chronology would be preferable to a duration derived from assumed sediment transport rates and modes. Our approach is designed only to bracket a range of possibilities and, while quite broad, we note that error bars even on a carbonate radiocarbon date from this period and setting would be unlikely to provide improvement on the few hundred years that our estimates bracket. For example, on the Coulman High 8.6 cal ka date from McKay et al 2016, they quote 1 sigma errors of +/- 242 (1 sigma range 484 years). Our sediment flux-based approach recognises the possibility that actual fluxes would vary significantly between different catchments; this is why we explore a range of values crossing orders of magnitude. In light of this we find it unlikely that our story would require a complete overhaul if/when a better chronology becomes available; dates that would push our calculated durations beyond the upper/lower limits calculated here would push sediment transport rates outside an envelope that already captures a range of different transport modes and geographic settings, and therefore would suggest something quite unusual occurring in this setting. However, our original ms clearly did not appropriately reflect uncertainties in our approach, or our treatment of those uncertainties, and we have carefully reviewed our wording in this revised version – particularly the Methods section.*

Besides that, the manuscript is very well written, making it easy and enjoyable to read. I like their story and would like to see it published after addressing my more general comments above and a few minor comments that I list below.

-> *Thanks!*

Minor comments:

Line 36: Capitalize "Last Glacial Maximum".

Done

Line 55: Avoid questions in the manuscript.

This is a matter of writing style only. If Nature Communications' preferred style is to avoid questions in the text then I will re-write, but otherwise prefer to leave it as originally written.

Line 62: Write "decades to centuries" instead of "10s-100s".

Done

Line 85: Replace "seascape" with "seafloor".

Done

Line 121: This only applies if moraines were formed at the same time. Is there core evidence (ages)? If not, phrase more carefully here.

There is no unequivocal core evidence (dates are too unreliable) but grounding line shape revealed by grounding line landforms clearly indicates 'unzipping' of bank and trough ice from all sides. Text reworded.

Line 161: Write "oriented perpendicular".

Done

Line 174: Write "ENE".

The flow is oriented initially E and then NE, so "E-to-NE" better captures this pattern than giving a single direction. Text is unchanged.

Line 182: Replace "ever" with "increasingly".

Done

Line 195: Write "the most distal wedge".

Done

Line 270: What is the Ross Sea ice sheet? Do you mean formerly grounded ice in the RSE?

Done

Replace "non-monotonic" with "heterogenic".

Done

Lines 288-301: Use the term "stillstand".

Done

Lines 319-321: Very nice and interesting result and nicely comparable to recent events at Larsen C (maybe cite).

References to dynamic effects of the loss of Larsen B are given earlier in the paragraph; consequences of Larsen C calving remain to be seen.

Figures:

Fig. 1: Give more regional context for Antarctic overview map, e.g. "WAIS", "EAIS", Amundsen Sea, Weddell Sea, ... just for reference. And maybe include general coordinates. Maybe adjust color range in main figure so that troughs in RSE become more pronounced and give color code in legend. Explain "T.A.M." in caption.

Added annotation to inset. Explained TAM in caption. Enhanced depth scale and added legend.

Fig. 2: Label "Drygalski Trough" in A. Again: also give more pronounced color range and give color code in legend. Maybe also give a legend with explained mapped landforms. F: I honestly cannot clearly see the crevasse-squeeze ridges. Give a zoom-in.

Labelled Drygalski Trough, strengthened bathymetry colour and added depth legend. In F, the crevasse-squeeze ridges are the most pronounced & irregularly shaped forms; label lines adjusted to point directly towards the ridge crests.

Fig. 3: Referring to this figure: Mention in text that further mapping just south of 76°S and just E and W of 164°E could be crucial in order to explain different MSGL orientations. It would be good to see the actual locations of sub-bottom profiles in C and D - maybe create a zoom-in of A as an inset.

Added comment regarding data needs in the caption. Inset added with profile lines for C&D.

Fig. 4: It would be good to see the locations of sub-bottom profiles on bathymetric maps and not only on the sediment thickness map.

Done

Fig. 7: Give direction of reconstructed flow lines and use darker color.

Done

Reviewers' comments:

Reviewer #1 (Remarks to the Author):

** What are the major claims of the paper? **

The major claims are 1) unzipping of the ice sheet grounding line caused an ice flow direction change and enabled a grounding line advance; and 2) they can quantify the scale and duration of this readvance, and the readvance was sustained for centuries.

The revised manuscript is a great improvement in how it is presented, and is less open to misinterpretations. I am mostly in agreement with the multibeam flow line interpretations (so Claim 1 is supported by the data), but I have some comments below that are intended to be constructive.

I still have outstanding concerns around the grounding zone duration calculations. However, they have significantly toned down this in the revised manuscript, and I endorse this paper being accepted with aspect included - if the caveats are more clearly explained in the main text, and the revised calculations using the Golledge et al., 2012 sediment flux estimates are removed.

** Claim 1: Constraining the extent of northward flowing ice.**

In my original review, I never argued was no northward flow in the region of Fig 3B (so there is no "hypothesis of southward-only flow"). However, it is a big assumption to infer from subtle (but conclusive) features highlighted to the north (Figure 3B), to what is happening in the region of the elliptical dotted line in Figure 3A. I think the dotted arrow in figure 3A is misleading regarding the significance of this event, and does not imply a complete 180 degree shift in flow direction along the entire axis of this trough. I just cannot see the evidence is strong enough to support this (although it is possible). The authors themselves state the directions in this trough are indeterminable (line 144).

In this context, it is ambiguous in the text as written about the spatial scale of this reversal event. As they note, Golledge et al., 2014 model a switch in David Glacier flow during the deglaciation, so it is physically plausible this happened, but this model also highlights the scale of this shift is rather localized, and the David Glacier is subject to period reconfigurations than are not always related directly to grounding retreat (see the Supplementary Movie file associated with that paper; some snapshots are provided in an attached file to this review).

Editorial Note: Attached screenshot included below.

Snapshots from Golledge et al., 2014: PNAS - sup info video

Screenshot of Supplementary Movie 1, taken from N. R. Golledge, L. Menviel, L. Carter, C. J. Fogwill, M. H. England, G. Cortese & R. H. Levy. Antarctic contribution to meltwater pulse 1A from reduced Southern Ocean overturning. *Nature Communications*. **5**, 5107 (2014).

The Golledge et al., 2014 model also shows a short readvance in this region following retreat in the central embayment – so there is also a clear physical basis behind this multibeam interpretation, albeit with parameterization and scale issues resulting in the exact timing and extent of modelled readvance being different.

While this consistency with models is well discussed in the manuscript, it does make me query some the interpretation in Figure 2A (and thus Figure 7). This criticism is intended to be constructive, but is there clear evidence for a continuous grounding line connecting grounded ice from the south of Ross Island in the Coulman High region (e.g. the retreat line drawn directly below the annotation “Central Basin” in figure 2A) with the lobe of ice coming out of Mawson/McKay glaciers? The manuscript is arguing this is a readvance of these outlets glaciers, following retreat (unzipping) in the deep troughs, but this doesn't require connecting to a central Ross Sea ice mass. This figure doesn't really capture this readvance concept clearly in my view.

Ice sheet models show this region can form a large, but discrete lobe of grounded ice that is independent of being connected to grounded ice to the south of Ross Island (see above figure for an example; but Pollard and Deconto et al., 2003 also show this). This has implications for the 8.6ka date recovered at Coulman High, which can remain in a calving line proximal position throughout the duration of this readvance? To me, the different flow directions (e.g. Figure 4B) in this region are potentially more important than those at the mouth of David Glacier regarding the reconfiguration leading to a readvance hypothesis presented – but this is not captured well in Figure 2A and is greatly oversimplified in Figure 7. I think this is also more consistent with the discussion argument around the mechanisms for this readvance (which is a nice discussion). If the authors agree, I think Figure 2A could have this retreat history interpretation removed, and revise the two discrete images in Figure 7? (similar to how Greenwood et al., 2012 presented multiple ice flow direction in their Figure 5).

In this context, I think they could reasonably argue the readvance was likely to have occurred sometime between 8.6 ka and the final retreat dates of ~6 ka along the Scott coast that are cited. This is consistent with the upper range of estimates derived from their sediment volume estimates.

Are they novel and will they be of interest to others in the community and the wider field? If the conclusions are not original, it would be helpful if you could provide relevant references.

Numerous previous studies have demonstrated changing ice flow directions during deglaciation, due to a range of processes (Dowdeswell et al., 2006; *Geology*; Graham et al., 2009 *QSR*; Halberstadt et al., 2016 in the Ross Sea). The point of difference here is that this reconfiguration of ice resulted in readvance of the grounding line by at least 50km (potentially unrelated to climate drivers), but probably greater than this. Given the importance of this region in interpreting Ross Sea ice sheet retreat (one of the largest contributors to post LGM retreat in the Antarctica), I believe this is sufficiently novel for publication in *Nature Communications*.

Is the work convincing, and if not, what further evidence would be required to strengthen the conclusions?

The multibeam data and figures are beautiful and very well presented, and I don't have any problems with the first order interpretations of these data – only relatively trivial concerns regarding the extent of the flow direction changes discussed above that I hope are helpful.

However, I am still very skeptical the constraints for duration of the grounding zone. I can accept

this is a valid conceptual argument to present, but the way it is currently written in the text makes it sound like they use an alternative approach to narrowed this down to 360-600 years using the potential erosion rates by Golledge et al. 2013.

While Golledge et al., do state the possibility of such an approach contributing to these sort of studies, it is being misused here. Golledge et al. 2013 initiated this discussion by saying it was only intended to be indicative of relative patterns. It is not appropriate to use the erosion potential cited in that paper at this scale, and as noted by Golledge et al., this value is potential erosion and not a constraint on absolute erosion. Specifically they cited "We can use the calculated driving stress and basal velocities as inputs to Equation (3) to calculate erosion potential at the base of the ice sheet (Pollard and DeConto, 2003), accepting that the large number of uncertainties surrounding this process means we are limited to determining a spatial pattern of potential erosion, rather than absolute values."

As such, using these absolute value at face value is a flawed approach, and does not refine this estimate robustly. The uncertainty caveats stated above are not propagated into these new GZW duration calculations.

The authors have discussed some of the caveats in the methods, but I feel this is undone by using the values present by Golledge et al., 2012 – which was clear that these values should not be taken at face value. In my view, they should only present the widest range of uncertainties, note these caveats in the main text, and lay down the challenge to improve the chronology from data in this region to refine our understanding of grounding zone wedge processes and formation times. The paper is strong enough without this be a main focus in my view.

On a more subjective note, do you feel that the paper will influence thinking in the field?

Yes. It provides excellent context to interpret future sediment cores and cosmogenic deflation data to understand the significance of deglacial chronologies in this critical area, and gives a very nice overview of mechanisms that could explain readvances of selected catchments during the last glacial termination in the Ross Sea.

Reviewer #3 (Remarks to the Author):

I am happy how the authors addressed my own but also the comments and concerns of the other reviewer. In its current state, I support a quick publication of the manuscript.

Kind regards,
J.P. Klages

Reviewer #1:

What are the major claims of the paper?

The major claims are 1) unzipping of the ice sheet grounding line caused an ice flow direction change and enabled a grounding line advance; and 2) they can quantify the scale and duration of this readvance, and the readvance was sustained for centuries.

The revised manuscript is a great improvement in how it is presented, and is less open to misinterpretations. I am mostly in agreement with the multibeam flow line interpretations (so Claim 1 is supported by the data), but I have some comments below that are intended to be constructive.

I still have outstanding concerns around the grounding zone duration calculations. However, they have significantly toned down this in the revised manuscript, and I endorse this paper being accepted with aspect included - if the caveats are more clearly explained in the main text, and the revised calculations using the Gollledge et al., 2012 sediment flux estimates are removed.

We're pleased our revisions are seen as improvements. We have addressed the relatively minor queries about our interpretations, detailed below.

Claim 1: Constraining the extent of northward flowing ice.

In my original review, I never argued was no northward flow in the region of Fig 3B (so there is no "hypothesis of southward-only flow"). However, it is a big assumption to infer from subtle (but conclusive) features highlighted to the north (Figure 3B), to what is happening in the region of the elliptical dotted line in Figure 3A. I think the dotted arrow in figure 3A is misleading regarding the significance of this event, and does not imply a complete 180 degree shift in flow direction along the entire axis of this trough. I just cannot see the evidence is strong enough to support this (although it is possible). The authors themselves state the directions in this trough are indeterminable (line 144).

*We in fact agree – the lineations within the dotted ellipse are ambiguous in their direction, and cannot support a full reversal of flow *along the entire axis of the trough*. The spatial scale of the reversal remains difficult to constrain, and we have not attempted to do so. If these central trough lineations are directed northward, then it means their source must come from at least south of Mawson; Mackay would realistically be the first possible TAM source (some ~70km south, or otherwise even further south). If they are directed south, then it means their source must come from well north of Mawson (at least 40km or so) and, given a slight NE-ward bend to their orientation, even sourcing from David glacier is looking a little suspect (would be more oblique to trough?) – perhaps Terra Nova Bay?*

*The main point that we wish to draw attention to regarding these lineations within the ellipse, is that they preclude simultaneous northward and southward flow, diverging from SVL outlet sources. They give a false impression of a continuous flowset; one which cannot, in practice, exist with both northward and southward flow at the north & south extremities of the assemblage. Somewhere in between, flow must have reversed, but this doesn't *necessarily* mean that the full length of southern Drygalski Trough experienced two separate flow events in entirely the opposite direction.*

-> We have edited the text around lines 146-152. (Changes are tracked in the revised Word document, and all line numbers quoted here refer to when tracked changes are displayed – they do not correspond to the submission system-built pdf, which does not display the tracking.)

In this context, it is ambiguous in the text as written about the spatial scale of this reversal event. As they note, Gollledge et al., 2014 model a switch in David Glacier flow during the deglaciation, so it is physically plausible this happened, but this model also highlights the scale of this shift is rather localized, and the David Glacier is subject to period reconfigurations than are not always related directly to grounding retreat (see the Supplementary Movie file associated with that paper; some snapshots are provided in an attached file to this review). The Gollledge et al., 2014 model also shows a short readvance in this region following retreat in the central embayment – so there is also a clear

physical basis behind this multibeam interpretation, albeit with parameterization and scale issues resulting in the exact timing and extent of modelled readvance being different.

Agree – I think the style of behaviour is instructive; the exact details shouldn't be taken for literal comparison with timing or spatial position.

While this consistency with models is well discussed in the manuscript, it does make me query some the interpretation in Figure 2A (and thus Figure 7). This criticism is intended to be constructive, but is there clear evidence for a continuous grounding line connecting grounded ice from the south of Ross Island in the Coulman High region (e.g. the retreat line drawn directly below the annotation "Central Basin" in figure 2A) with the lobe of ice coming out of Mawson/McKay glaciers? The manuscript is arguing this is a readvance of these outlets glaciers, following retreat (unzipping) in the deep troughs, but this doesn't require connecting to a central Ross Sea ice mass. This figure doesn't really capture this readvance concept clearly in my view.

No, the data do not require a continuous grounding line as described. It was the intention with these retreat lines to depict a very generalised pattern of retreat (e.g. as if from a scale that doesn't consider all the details), and then to delve into the details in subsequent figures. However, you're correct, it gives an impression that the two sectors remain connected, whereas that doesn't have to be the case. Our data and our reconstructed readvance do not require grounded ice to still be anywhere close to eastern Ross Island (though they also do not refute this).

-> We have broken two of these schematic retreat margin lines in Fig. 2A, so as not to imply continuity that might not be borne out.

Ice sheet models show this region can form a large, but discrete lobe of grounded ice that is independent of being connected to grounded ice to the south of Ross Island (see above figure for an example; but Pollard and Deconto et al., 2003 also show this). This has implications for the 8.6ka date recovered at Coulman High, which can remain in a calving line proximal position throughout the duration of this readvance?

Yes, in principle this site could remain ice free during this episode, if that could be reconciled with the Coulman High core lithostratigraphy – we have no conclusive landform evidence of ice cover (either grounded or shelf ice).

To me, the different flow directions (e.g. Figure 4B) in this region are potentially more important than those at the mouth of David Glacier regarding the reconfiguration leading to a readvance hypothesis presented – but this is not captured well in Figure 2A and is greatly oversimplified in Figure 7.

Fig. 4B doesn't really show flow direction – perhaps you mean Fig. 4C? Fig. 4C shows the regional flow directions, including the area around eastern Ross Island referred to above.

The question of a shift in flow direction revealed by lineations around Ross Island is a good point. The orientation of lineations and of ice-marginal landforms at the eastern end of Ross Island is in distinct contrast to those linked to the readvance lobe from Mackay/Mawson direction. This, together with the consistency of moraine orientation throughout Central Basin, is quite a persuasive argument for all of this assemblage denoting the first retreat, prior to readvance, and for that subsequent readvance to have a different source (from the west) that erases parts of the earlier retreat to the south.

-> We have commented further on this on lines 201-205 and edited Fig. 4C to illustrate the lineation conflict explicitly: flowsets fs1 vs fs3.

The relative timing of the landforms east of Ross Island is the key uncertainty in this question. There are fragmentary GZWs with slightly ambiguous orientations lying east of Beaufort Island, which could possibly link a Ross Island grounding line to the Mackay/Mawson flowset. Such a grounding line shape/position could be consistent with i) a first retreat, with the southern R.I. part

then pinned while the SVL part readvanced; ii) connected retreat after a more spatially limited readvance, predominantly driven by SVL outlets but with a GL still connected to the main Ross Sea supply; or iii) these wedges are not connected at all, and different marginal landforms here represent different relative timings. In none of these scenarios can we say that the GL detached further south of RI – we have no data to either confirm or reject that hypothesis. These remaining uncertainties are the reason we limit our interpretation of readvance magnitude to the conservative estimate of 50km – the distance for which we have conclusive evidence (direct overprinting).

I think this is also more consistent with the discussion argument around the mechanisms for this readvance (which is a nice discussion). If the authors agree, I think Figure 2A could have this retreat history interpretation removed, and revise the two discrete images in Figure 7? (similar to how Greenwood et al., 2012 presented multiple ice flow direction in their Figure 5).

-> In Fig. 2, we have broken two of the schematic retreat margin lines, so as not to imply continuity that might not be borne out.

-> In Fig. 4, we have added flowset outlines to indicate the complexity in flow directions around Ross Island. These clearly show the potential for truncation (flow reconfiguration linked to the readvance), but since there is no direct superimposition, the magnitude of readvance and the geometry of the pre-readvance retreat remain uncertain. Adding flowset outlines (in the style of Greenwood et al 2012) to Fig. 4C was preferable to Fig. 7, as the 'raw mapping' already in Fig. 4C provides the underlying basis for those summary groups.

-> In Fig. 7, our uncertainties over the pre-readvance flow and grounding line configuration, and the behaviour of Central Basin ice during the readvance event, are made clearer in both annotation and in caption.

In this context, I think they could reasonably argue the readvance was likely to have occurred sometime between 8.6 ka and the final retreat dates of ~6 ka along the Scott coast that are cited. This is consistent with the upper range of estimates derived from their sediment volume estimates.

We think this is a reasonable (favourable?) interpretation, while we recognise there may be other possible solutions that more (and more robust) dates could help to resolve.

Are they novel and will they be of interest to others in the community and the wider field? If the conclusions are not original, it would be helpful if you could provide relevant references.

Numerous previous studies have demonstrated changing ice flow directions during deglaciation, due to a range of processes (Dowdeswell et al., 2006; Geology; Graham et al., 2009 QSR; Halberstadt et al., 2016 in the Ross Sea). The point of difference here is that this reconfiguration of ice resulted in readvance of the grounding line by at least 50km (potentially unrelated to climate drivers), but probably greater than this. Given the importance of this region in interpreting Ross Sea ice sheet retreat (one of the largest contributors to post LGM retreat in the Antarctica), I believe this is sufficiently novel for publication in Nature Communications.

Thanks for summarising – we agree! We consider it significant, as a general style of behaviour, that a response to regional margin retreat might be sustained (albeit time-limited) margin advance.

Is the work convincing, and if not, what further evidence would be required to strengthen the conclusions?

The multibeam data and figures are beautiful and very well presented, and I don't have any problems with the first order interpretations of these data – only relatively trivial concerns regarding the extent of the flow direction changes discussed above that I hope are helpful.

However, I am still very skeptical the constraints for duration of the grounding zone. I can accept this is a valid conceptual argument to present, but the way it is currently written in the text makes it sound like they use an alternative approach to narrowed this down to 360-600 years using the potential erosion rates by Golledge et al. 2013. While Golledge et al., do state the possibility of such an approach contributing to these sort of studies, it is being misused here. Golledge et al. 2013 initiated this discussion by saying it was only intended to be indicative of relative patterns. It is not appropriate to use the erosion potential cited in that paper at this scale, and as noted by Golledge et al., this value is potential erosion and not a constraint on absolute erosion. Specifically they cited "We can use the calculated driving stress and basal velocities as inputs to Equation (3) to calculate erosion potential at the base of the ice sheet (Pollard and DeConto, 2003), accepting that the large number of uncertainties surrounding this process means we are limited to determining a spatial pattern of potential erosion, rather than absolute values."

The 'narrowing down' was in fact unintended, but I agree, the revised text and results now misrepresent the uncertainties in both approaches. The intent was to explore an alternative, semi-independent approach, to examine the robustness of the original one and whether this sector of the Ross Sea fits within the range of values that we've adopted from the literature from other locations - i.e. a question of representativeness, to an order of magnitude. The numbers ended up, at face value, giving us a narrower range. But I agree, we've lost the uncertainties (that Golledge et al do clearly flag) along the way.

-> We have removed this aspect from the main text, methods and supplementary material, satisfied that an alternative approach gives us a correct order of magnitude but that it's inappropriate to draw on this to narrow our results.

-> We quote the wider range for our GZW complex formation time (100-1600 years), and update our estimate of grounding line discharge (Supplm Table 2) accordingly.

As such, using these absolute value at face value is a flawed approach, and does not refine this estimate robustly. The uncertainty caveats stated above are not propagated into these new GZW duration calculations. The authors have discussed some of the caveats in the methods, but I feel this is undone by using the values present by Golledge et al., 2012 – which was clear that these values should not be taken at face value. In my view, they should only present the widest range of uncertainties, note these caveats in the main text, and lay down the challenge to improve the chronology from data in this region to refine our understanding of grounding zone wedge processes and formation times. The paper is strong enough without this be a main focus in my view.

-> We present GZW formation time according to the range yielded by the original sediment flux approach.

-> We add a comment to the Chronology section regarding the challenges of refining time constraints from both an absolute chronology and sediment flux perspective (lines 310-13).

On a more subjective note, do you feel that the paper will influence thinking in the field?

Yes. It provides excellent context to interpret future sediment cores and cosmogenic deflation data to understand the significance of deglacial chronologies in this critical area, and gives a very nice overview of mechanisms that could explain readvances of selected catchments during the last glacial termination in the Ross Sea.

Reviewer #3:

I am happy how the authors addressed my own but also the comments and concerns of the other reviewer. In its current state, I support a quick publication of the manuscript.

Kind regards,
J.P. Klages

We're pleased the reviewer is satisfied with our revisions.